# How mindfulness, self-compassion, and experiential avoidance are related to perceived stress in a sample of university students

David Martínez-Rubio[1,2☯], Ariadna Colomer-Carbonell[1,3,4,5☯], Juan P. Sanabria-Mazo[1,3,4,5]*, Adrián Pérez-Aranda[6,7], Jaime Navarrete[1,3,4,5]*, Cristina Martínez-Brotóns[1,2], Cristina Escamilla[1], Anna Muro[6], Jesús Montero-Marín[1,3,4,5,8‡], Juan V. Luciano[3,4,5,9‡], Albert Feliu-Soler[5,9‡]

1 Faculty of Health Sciences, Department of Psychology, Universidad Europea de Valencia, Valencia, Spain, 2 Psicoforma, Integral Psychology Center, Valencia, Spain, 3 Institut de Recerca Sant Joan de Déu, Esplugues de Llobregat, Spain, 4 Teaching, Research & Innovation Unit, Parc Sanitari Sant Joan de Déu, St. Boi de Llobregat, Spain, 5 CIBER of Epidemiology and Public Health (CIBERESP), Madrid, Spain, 6 Faculty of Psychology, Department of Basic, Developmental, and Educational Psychology, Autonomous University of Barcelona, Cerdanyola del Vallès, Spain, 7 Aragon Institute for Health Research, IIS Aragon, Zaragoza, Spain, 8 Department of Psychiatry, Warneford Hospital, University of Oxford, Oxford, United Kingdom, 9 Department of Clinical and Health Psychology, Autonomous University of Barcelona, Cerdanyola del Vallès, Spain

☯ These authors contributed equally to this work.
‡ JM-M, JVL and AF-S are Joint senior co-authorship
* juanpablo.sanabria@sjd.es (JPS-M); jaime.navarrete@sjd.es (JN)

## Abstract

University students constitute a population that is highly vulnerable to developing mental health problems, such as distress. The role of different variables associated with the development of states of stress has been studied in order to identify potential risk and protective factors. This study explored whether mindfulness, self-compassion, and experiential avoidance, while controlling for specific sociodemographic and academic variables, were potential significant protective or risk factors explaining perceived stress in a sample of 589 Spanish university students (81.2% female, age range 18–48 years). A hierarchical multiple regression analysis was performed using an exploratory cross-sectional design. Higher experiential avoidance, lower self-compassion, lower mindfulness, not perceiving family support, higher total study hours per week, having a partner (vs. being single), being female (vs. being male), and being older were significantly associated with higher levels of perceived stress. In conclusion, perceived stress in our sample was positively associated with experiential avoidance, which could be regarded as a potential psychological risk variable. In contrast, perceived stress was negatively correlated with self-compassion and mindfulness, which, in turn, could be seen as protective factors. Accordingly, it is concluded that programmes aimed at reducing stress and at improving well-being among university students should include experiential avoidance, self-compassion, and mindfulness as therapeutic targets.

**Data Availability Statement:** For transparency and analytical reproducibility purposes, SPSS data can be accessed at OSF: https://osf.io/qjp5h/.

**Funding:** This study was financially supported by AGAUR in the form of a FI predoctoral contract (FI_B/00216) awarded to AC-C. This study was also financially supported by the Institute of Health Carlos III in the form of a PFIS predoctoral contract (ISCIII; FI20/00034) awarded to JPS-M. This study was also financially supported by ISCIII in the form of a "Sara Borrell" postdoctoral contract (CD20/00181) awarded to AP-A, a research contract (ICI20/00080) awarded to JN, and a "Miguel Servet" research contract (CP21/00080) awarded to JM-M. This study was also financially supported by the Serra Húnter program in the form of a grants awarded to AF-S (UAB-LE-8015) and AM (UAB-LE-603). This study was also financially supported by CIBER of Epidemiology and Public Health (CIBERESP CB22/02/00052; ISCIII). The funders had no role in study design, data collection and analysis, decision to publish, or preparation of the manuscript.

**Competing interests:** The authors have declared that no competing interests exist.

## Introduction

The mental health of university students has become a worldwide concern [1]. It is estimated that around 35% of university students meet the diagnostic criteria for a mental disorder [2]. In fact, their risk for developing a mental disorder has significantly increased up to around 50% as a result of the COVID-19 pandemic [3,4]. Overall, university students are particularly vulnerable to stress, which in this population is strongly related to academic underperformance, i.e., failure to fulfil academic obligations, and problematic health behaviors, such as substance use [5,6]. In addition, higher perceived stress levels in university students are associated with poorer quality of life, well-being, and sleep quality [7].

Given the high prevalence of stress among university students, there has been a burgeoning interest in detecting potential protective skills that could be enhanced to improve their mental health, such as mindfulness, self-compassion, and psychological flexibility [8,9]. Many studies have consistently indicated the negative association of mindfulness, compassion, and psychological flexibility with perceived stress in general populations [10–12]. Below is a brief conceptualization of them and a summary of the research studying its association with perceived stress among university students.

Mindfulness can be defined as a two-component mood that involves the self-regulation of attention, i.e., paying attention to the immediate experience (including internal experiences such as thoughts), and a curious, open, and accepting orientation towards it [13]. The cultivation of mindfulness skills leads to improvements in attentional control, acceptance of one's experience, and non-reactivity to acute stressors [14,15]. Thus, high-level mindfulness individuals frequently respond to acute stressors with non-reactive acceptance instead of showing a pattern of excessive and blunted responses that could lead to a host of poor health outcomes [16]. This is reflected in previous studies involving university students. For instance, Burger et al. [17] investigated the associations between psychological distress and mindfulness in a sample of 174 university students and found that higher levels of mindfulness (especially the facet of acting with awareness) significantly predicted lower levels of negative affect, fatigue, nervousness, and agitation. Similarly, Muro et al. [18] found that mindfulness was associated with higher levels of life satisfaction in a sample of Spanish university students, a result which is consistent with the multicentre cross-sectional study of Salvarani et al. [19], who found that undergraduate nursing students ($n = 622$) with higher dispositional mindfulness scores had lower levels of stress symptoms.

Compassion is a cognitive, affective, and behavioural process that involves recognising suffering, common humanity, empathy, tolerance for uncomfortable feelings, and motivation to act or acting to alleviate the suffering of others and/or oneself [20]. Being self-compassionate during stressful life circumstances enables a pool of emotion-regulation strategies (i.e., healthy reappraisals, emotional acceptance, and self-soothing) that protects against distress [21]. In this line, Rahmandani et al. [22] found a significant negative association between self-compassion and several dimensions of distress (e.g., loss of confidence, anxiety), suggesting that high self-compassion might explain a lower level of stress in university students. In addition, Chan et al. [23] have found in a sample of 536 university students that compassion for others and self-compassion were linked to lower levels of perceived stress.

Finally, psychological flexibility is defined as the ability to consciously make contact with the present moment and behave in a way that serves valued goals [24]. Previous research about this construct have focused on experiential avoidance instead, which is a process aimed at trying to facilitate psychological flexibility. Experiential avoidance is defined as a "phenomenon that occurs when a person is unwilling to remain in contact with particular private experiences (e.g., bodily sensations, emotions, thoughts, memories, behavioural predispositions) and takes

steps to alter the form or frequency of these events and the contexts that occasion them" ([24], p. 1154). Individuals who tend to rigidly avoid uncomfortable or unwanted internal experiences are more vulnerable to stress because this avoidant response pattern inhibits effective behaviors to cope with stress and might even increase the frequency of these unwanted internal experiences [25]. For instance, Chou et al. [26] found in a sample of 500 university students that experiential avoidance was a significant predictor of Internet addiction, significant depression, and suicidality. Moreover, Farr et al. [27] found that experiential avoidance was a significant predictor of depression, anxiety, and stress, this positive association being moderated by self-compassion. Specifically, these authors showed that individuals high in self-compassion reported lower levels of depressive symptoms across low to high experiential avoidance levels than individuals with low self-compassion levels [27].

Mindfulness, self-compassion, and experiential avoidance are three interrelated processes. Both compassion for others and self-compassion have consistently been found to be positively related to trait mindfulness (e.g. [28]). In addition, experiential avoidance and self-compassion are strongly and negatively associated [29]. Similarly, mindfulness has been found to negatively correlate with experiential avoidance [30]. According to Neff [31], there is a reciprocal association between mindfulness and self-compassion, in which each one is necessary for facilitating and enhancing the other. For example, a self-compassionate response requires a mindful approach to painful thoughts and feelings. However, an important distinction between them is that mindfulness is applied to pleasant, neutral, or unpleasant experiences, whereas self-compassion is focused on suffering. Moreover, self-compassion is focused on the global self, meanwhile mindfulness can be applied to the self as well as to thoughts, feelings, and sensations [32]. Furthermore, being unjudgmental and acceptance (central facets of mindfulness) can be considered the beneficial counterparts of experiential avoidance [30]. Likewise, mindfulness and experiential avoidance are drawn from very different theoretical contexts, i.e., Eastern philosophy and functional contextualism, respectively.

Overall, these are three psychological constructs of "third wave" cognitive behavioral therapies, which focus on the relationship between the individual and his/her thoughts and emotions regardless of the content [12]. As mentioned, university students show increased risk for distress and focusing on specifically how they respond to the internal experiences (thoughts, emotions, urges, sensations, memories, etc.) associated with their stressful circumstances might provide further insights about the relevance of applying third-wave approaches (e.g., mindfulness-, compassion-based interventions or acceptance and commitment therapy). As a matter of fact, it seems that increasing mindfulness facets, compassion, or psychological flexibility (by reducing experiential avoidance) may help to improve psychological distress in university students (e.g. [17,19,27]). However, very few studies have simultaneously studied the ability of trait mindfulness, compassion, and experiential avoidance, along with sociodemographic and academic variables, to explain levels of psychological distress in university students. Including all those variables in the same model would add specific information about which of these psychological constructs should be more emphasized in mental health interventions for university students.

In this sense, Martínez-Rubio et al. [33] showed that experiential avoidance, self-compassion and the mindfulness facets of observing and acting with awareness were significantly related to lower burnout symptom ('overload', 'lack of development', and 'neglect') levels in a sample of 644 undergraduate nursing and psychology students, but the potential influence of sociodemographic and academic factors in that association was not controlled. Results showed that experiential avoidance was the strongest predictor of the burnout dimensions of 'overload' and 'lack of development', meanwhile the dimension of 'neglect' was better explained by acting with awareness [33].

Therefore, the aim of the current study was to explore the relative contributions of three processes that are therapeutically relevant to perceived stress (i.e., trait mindfulness, self-compassion, and experiential avoidance), a critical construct among university students who are at risk of burnout, depression, and anxiety. According to the literature, we expected that mindfulness, self-compassion, and experiential avoidance would explain a significant proportion of the variance of perceived stress after controlling for relevant sociodemographic and academic variables such as age, gender, or having a scholarship. The direction of the relationship between mindfulness/self-compassion and perceived stress was expected to be negative (inverse). Alternatively, a positive relationship was expected between experiential avoidance and stress. No prior hypotheses were established on the strength of the associations due to the scarce number of previous studies.

## Materials and methods

### Participants

Table 1 displays all the sociodemographic, academic, and psychological characteristics for the total sample. The study sample was composed of 589 university teaching students from the San Vicente Mártir Catholic University of Valencia (Spain), of whom 81.2% were female, and the mean age was 22.13 years old (SD = 3.90; range: 18–48). All participants were White. The inclusion criteria were: (1) being ≥18 years old, (2) being an undergraduate student in the field of education, and (3) being able to understand written Spanish.

### Instruments

The following sociodemographic characteristics were recorded using an ad hoc questionnaire (response options): age, gender (male/female), having a partner (yes/no), number of children, perceived presence of family support (insufficient/good/very good), employment (yes/no), and emancipation status (yes/no). Likewise, the following academic variables were recorded: having a scholarship (yes/no), which academic year the student was in, total study hours per week, and number of failed subjects over the previous examination period. Along with this survey, participants completed the aspects described below as part of a pencil-and-paper or online set of measurements.

The *Perceived Stress Questionnaire-Short Form* (PSQ-SF; [34]) is a 24-item scale that assesses perceived stress during the previous 30 days. The response format is a 4-point scale (from 1 = "almost never" to 4 = "almost always"). This questionnaire was validated for Spanish university students and showed adequate internal consistency and construct validity [35]. The total score ranges from 0 to 1 and can be calculated by adding all items, subtracting 24, and dividing by 72 [35]. Higher scores represent greater levels of perceived stress. The PSQ has demonstrated excellent internal consistency in the present sample with a Cronbach's α value of 0.93.

The *Five Facets Mindfulness Questionnaire-Short Form* (FFMQ-SF; [36]) is a 20-item questionnaire that measures the five facets of mindfulness: observing, describing, acting with awareness, non-judging of inner experience, and non-reactivity to inner experience. The response format is a 5-point scale (1 = "never or very rarely true", 5 = "very often or always true"). Tran et al. [36] validated the German and the Spanish version of this FFMQ-SF. Two subscales (describing and non-reactivity to inner experience) demonstrated inadequate internal consistency (Cronbach's α < .70), thus only the total score, which was computed by calculating the sum of all items, was retained for the analyses. Higher scores represent greater levels of mindfulness, with the total scores ranging from 20 to 100 points. In the current sample, the total score showed an adequate internal consistency with a Cronbach's α value of 0.76.

**Table 1. Sociodemographic, academic, and psychological characteristics of the sample (n = 589).**

| Sociodemographic variables | | | |
|---|---|---|---|
| Age (years), *M (SD)* | 22.13 (3.90) | | |
| Gender, *n (%)* | | | |
| Male | 110 (18.8) | | |
| Female | 475 (81.2) | | |
| Stable relationship, n (%) | | | |
| Yes | 304 (52.0) | | |
| No | 281 (48.0) | | |
| Children, *n (%)* | | | |
| None | 578 (98.3) | | |
| One or more | 10 (1.7) | | |
| Job, *n (%)* | | | |
| Yes | 102 (17.6) | | |
| No | 479 (82.4) | | |
| Left home, *n (%)* | | | |
| Yes | 112 (19.0) | | |
| No | 476 (81.0) | | |
| Family support, *n (%)* | | | |
| Yes, good/very good | 562 (95.4) | | |
| No, insufficient | 27 (4.6) | | |
| **Academic variables** | | | |
| Scholarship, *n (%)* | | | |
| Yes | 162 (27.6) | | |
| No | 426 (72.4) | | |
| Academic year, *n (%)* | | | |
| 1st | 120 (20.4) | | |
| 2nd | 146 (24.8) | | |
| 3rd | 123 (20.9) | | |
| 4th | 157 (26.7) | | |
| 5th | 42 (7.1) | | |
| Study hours per week, *M (SD)* | 34.92 (10.14) | | |
| Number of failed subjects, *M (SD)* | 0.54 (0.96) | | |
| **Psychological variables** | *M (SD)* | *Minimum* | *Maximum* |
| PSQ-SF (0–1) | 0.42 (0.17) | 0.01 | 0.97 |
| FFMQ-SF (20–100) | 63.70 (9.52) | 25 | 97 |
| SCS-SF (12–60) | 36.74 (8.51) | 12 | 60 |
| AAQ-II (7–49) | 22.85 (9.11) | 7 | 48 |

*Note. n* = frequencies; % = percentages; *M* = mean; *SD* = standard deviation; PSQ-SF = Perceived Stress Questionnaire-Short Form; FFMQ-SF = Five Facets Mindfulness Questionnaire-Short Form; SCS-SF = Self-Compassion Scale-Short Form; AAQ-II = The Acceptance and Action Questionnaire-II. Range of possible scores are shown in brackets.

 The *Self-Compassion Scale-Short Form* (SCS-SF; [37]) is a 12-item scale that provides an overview of how people might typically respond to themselves during times of struggle. This scale assesses six aspects related to self-compassion that are organised by pairs of opposing constructs: self-kindness-self-judgment, common humanity-isolation, and mindfulness-over-identification. The response format is on a 5-point scale (from 1 = "almost never" to 5 = "almost always"). A total score can be computed by calculating the sum of all items after

reversing those that are negatively worded (scores ranging from 12 to 60). Higher scores indicate more tendency towards self-compassion. The Spanish adaptation of the SCS has proved to be valid for the evaluation of self-compassion among the general population [38]. The SCS-SF showed an appropriate internal consistency in the present sample with a Cronbach's α value of 0.81.

The *Acceptance and Action Questionnaire-II* (AAQ-II; [39]) is a 7-item self-report that measures experiential avoidance. Responses are based on a 7-point Likert-type scale (from 1 = "never true" to 7 = "always true"). A total score can be calculated by adding up all items, with scores ranging from 7 to 49. Higher scores indicate a higher tendency to present experiential avoidance. The Spanish version of the AAQ-II was used, which has shown strong psychometric properties [40]. The AAQ-II demonstrated adequate internal consistency in the present sample with a Cronbach's α value of 0.89.

## Procedure

A cross-sectional self-report design was used, with a convenience sample of university students studying education at the San Vicente Mártir Catholic University of Valencia, Spain. The study was conducted according to domestic and international ethical standards (Helsinki and Tokyo Conventions) and it was approved by the Research Ethics Committee of the Catholic University of Valencia (registry number: H1455835241950; 2016).

Students were invited to participate at the end of their classes. They were informed about the voluntary nature of participation and the confidentiality of the data. In particular, it was explained to them that the aim of the study was to investigate the association between their levels of perceived stress, mindfulness, self-compassion, and experiential avoidance. After that, we obtained written informed consent from all students interested in the study. Then, they were asked to individually answer the study measurements through a pencil-and-paper or online survey (Survey Monkey; 46% of the cases). The survey response time was approximately 20 minutes. There was no payment for participating. Returning a blank survey was accepted without any associated punishment. The survey responses were collected from March 2016 to May 2016. Data were managed anonymously and were only used for the purposes of the study. Thus, the confidentiality of the participants included in the study, as well as their rights to withdraw their own personal information, was guaranteed.

## Data analysis

All data analyses were carried out using the IBM Statistical Package for the Social Sciences (SPSS) v26, Chicago, IL. A descriptive analysis of participant characteristics was carried out using means (*M*) and standard deviations (*SD*), for the continuous variables, and frequencies (*n*) and percentages (%), for the categorical variables. The internal consistency of the instruments was established by calculating Cronbach's alpha (α). Coefficients above 0.70 were considered adequate [41].

A hierarchical multiple regression was carried out to assess the ability of FFMQ-SF (mindfulness), SCS-SF (self-compassion), and AAQ-II (experiential avoidance) measurements to explain PSQ-SF (perceived stress) scores after controlling for the influence of sociodemographic and academic variables. Preliminary analyses were conducted to check the assumptions of multicollinearity, outliers, normality, linearity, and homoscedasticity (S1 and S2 Tables and S1 Fig), with no major violations noted. First, all categorical sociodemographic and academic variables were recoded as dummy variables as follows: 'gender' (0 = male, 1 = female), 'having a partner' (0 = no, 1 = yes), 'children' (0 = no, 1 = yes), 'perceived presence of family support' (0 = no = insufficient, 1 = yes = good/very good), 'employment' (0 = no, 1 = yes), 'left

home' (0 = no, 1 = yes), 'having a scholarship' (0 = no, 1 = yes), 'which academic year the student was in' (each academic year recoded as dummy variable; 0 = no, 1 = yes). These categorical variables along with 'age', 'total study hours per week', and 'number of failed subjects over the previous examination period' were entered at Step 1. Additionally, FFMQ-SF, SCS-SF, and AAQ-II scores were entered at Step 2.

Given the large number of predictors included in the hierarchical multiple regression, we adjusted for multiple comparisons using the Benjamini–Hochberg procedure [42] utilizing a false discovery rate of 0.05. In addition, a sensitivity analysis using GPower v3.1 was conducted to indicate the minimum effect size that the study was powered to detect. Pairwise deletion was the technique used to handle missing data. Semi-partial correlation coefficients (sr) were examined to get an indication of the unique contribution of each variable to the total R square. Specifically, if this coefficient is squared, the percentage of unique variance explained by an independent variable in the dependent variable is obtained [43].

Continuing along the path laid down by previous studies [44], the hierarchical multiple regression was re-tested including an ad hoc SCS-SF total score for which the mindfulness and overidentification scores were removed (S3 Table). This analysis was performed due to the partial overlap between trait mindfulness and self-compassion measurements (i.e., FFMQ-SF and SCS-SF, respectively; [32]).

## Results

Table 2 shows the summary of the results from the hierarchical multiple regression analysis. The complete output is shown in S2 Table. According to the sample size ($n$ = 589) and number of predictors (17 variables), the regression would be sensitive to effects of $f^2$ = 0.03 (small) with 80% power (alpha = .05, two-tailed). Sociodemographic and academic variables were entered at Step 1, explaining 12% of the variance in PSQ-SF scores ($f^2$ = 0.14). After entering the FFMQ-SF, SCS-SF, and AAQ-II scores at Step 2 the total variance explained by the model was 50%, $F(17, 542)$ = 32.29, $p < .001$, $f^2$ = 1. In other words, the psychological measures (FFMQ-SF, SCS-SF, and AAQ-II) explained an additional 38% in perceived stress levels (PSQ) after controlling for sociodemographic and academic variables, $R$ squared change = .381, $F$ change (3, 542) = 139.41, $p < .001$.

Being a woman ($sr$ = 0.16, $p < .001$), perceived family support ($sr$ = -0.21, $p < .001$), left home ($sr$ = 0.08, $p$ = .043), and study hours per week ($sr$ = 0.12, $p$ = .004) were significant predictors at Step 1. In the final model, among the psychological measurements, only SCS-SF and AAQ-II significantly explained the PSQ-SF scores, with the AAQ-II recording a higher semi-partial correlation value ($sr$ = 0.31, $p < .001$) than the SCS ($sr$ = -0.17, $p < .001$). That is, the AAQ-II and SCS-SF scores uniquely explained around 10% and 3%, respectively, of the variance in PSQ-SF scores. In addition, perceived family support ($sr$ = -0.16, $p < .001$), study hours per week ($sr$ = 0.12, $p < .001$), having a partner ($sr$ = 0.09, $p$ = .004), being female ($sr$ = 0.08, $p$ = .010), and age ($sr$ = 0.07, $p$ = .015) made a unique statistically significant contribution to explaining PSQ-SF scores. When the Benjamini–Hochberg correction was applied to correct for multiple comparisons, all significant effects remained significant.

When the hierarchical multiple regression was re-tested with the SCS-SF total score calculated without adding the mindfulness and overidentification items scores (S3 Table), FFMQ-SF did significantly explain PSQ-SF scores ($sr$ = -0.07; $p$ = .015), along with the aforementioned variables. When the Benjamini–Hochberg correction was applied to correct for multiple comparisons, all significant effects remained significant except for age (adjusted $p$ = .056). In particular, AAQ-II, SCS-SF, and FFMQ-SF scores uniquely explained around 11%, 2%, and 0.55%, respectively, of the variance in PSQ-SF scores.

**Table 2. Hierarchical multiple regression analysis summary predicting total perceived stress scores with sociodemographic, academic, and psychological variables.**

| Step and predictor variable | B | SE B | β | sr | R² | ΔR² |
|---|---|---|---|---|---|---|
| **Step 1** | | | | | .12 | **.12** |
| Constant | 0.40 | 0.08 | | | | |
| Age | 0.00 | 0.00 | .03 | 0.03 | | |
| Gender (0 = male; 1 = female) | 0.07 | 0.02 | .16*** | 0.16 | | |
| Having a partner (0 = no, 1 = yes) | -0.01 | 0.01 | -.02 | -0.02 | | |
| Children (0 = no, 1 = yes) | 0.01 | 0.06 | .01 | 0.01 | | |
| Perceived family support (0 = no, 1 = yes) | -0.18 | 0.04 | -.22*** | -0.21 | | |
| Employment (0 = no, 1 = yes) | 0.04 | 0.02 | .08 | 0.08 | | |
| Left home (0 = no, 1 = yes) | 0.04 | 0.02 | .09* | 0.08 | | |
| Scholarship (0 = no, 1 = yes) | 0.0 | 0.02 | .01 | 0.01 | | |
| Being in their 1st academic year (0 = no, 1 = yes) | 0.00 | 0.02 | .00 | 0.00 | | |
| Being in their 2nd academic year (0 = no, 1 = yes) | 0.04 | 0.02 | .09 | 0.07 | | |
| Being in their 3rd academic year (0 = no, 1 = yes) | 0.02 | 0.02 | .05 | 0.04 | | |
| Being in their 5th academic year (0 = no, 1 = yes) | 0.01 | 0.03 | .01 | 0.01 | | |
| Study hours per week | 0.00 | 0.00 | .12** | 0.12 | | |
| Number of failed subjects | 0.01 | 0.00 | .02 | 0.02 | | |
| **Step 2** | | | | | .50 | **.38** |
| Constant | 0.39 | 0.08 | | | | |
| Age | 0.00 | 0.00 | .10* | 0.07 | | |
| Gender (0 = male; 1 = female) | 0.04 | 0.01 | .08* | 0.08 | | |
| Having a partner (0 = no, 1 = yes) | 0.03 | 0.01 | .09** | 0.09 | | |
| Children (0 = no, 1 = yes) | -0.00 | 0.04 | -.00 | -0.00 | | |
| Perceived family support (0 = no, 1 = yes) | -0.14 | 0.03 | -.17*** | -0.16 | | |
| Employment (0 = no, 1 = yes) | 0.03 | 0.01 | .06 | 0.06 | | |
| Left home (0 = no, 1 = yes) | 0.02 | 0.01 | .05 | 0.05 | | |
| Scholarship (0 = no, 1 = yes) | 0.00 | 0.01 | .01 | 0.01 | | |
| Being in their 1st academic year (0 = no, 1 = yes) | -0.01 | 0.02 | -.02 | -0.01 | | |
| Being in their 2nd academic year (0 = no, 1 = yes) | 0.03 | 0.02 | .07 | 0.05 | | |
| Being in their 3rd academic year (0 = no, 1 = yes) | 0.00 | 0.02 | .01 | 0.01 | | |
| Being in their 5th academic year (0 = no, 1 = yes) | 0.01 | 0.02 | .01 | 0.01 | | |
| Study hours per week | 0.00 | 0.00 | .13*** | 0.12 | | |
| Number of failed subjects | 0.00 | 0.00 | .02 | 0.02 | | |
| FFMQ-SF | -0.00 | 0.00 | -.07 | -0.06 | | |
| SCS-SF | -0.01 | 0.00 | -.23*** | -0.17 | | |
| AAQ-II | 0.01 | 0.00 | .43*** | 0.31 | | |

*Note*. The dependent variable was PSQ-SF scores. Variable 'Being in their 4th academic year' was excluded by SPSS from the models because it showed impossible tolerance values. B = unstandardised beta values; SE B = standard error of B; β = standardised beta values; sr = semi-partial correlation coefficient; R² = coefficient of determination; ΔR² = coefficient of determination change; FFMQ-SF = Five Facets Mindfulness Questionnaire-Short Form; SCS-SF = Self-Compassion Scale-Short Form; AAQ-II = Acceptance and Action Questionnaire-II.

* $p < .05$

** $p < .01$

*** $p < .001$.

## Discussion

This study examined the explanatory power of mindfulness, self-compassion, and experiential avoidance with regards to perceived stress in a sample of university students while controlling

for the influence of relevant sociodemographic and academic variables. When all of them were taken into consideration as potential explanatory variables of perceived stress in the regression model, mindfulness, self-compassion, and experiential avoidance explained a greater variance of perceived stress than all sociodemographic and academic variables together.

In particular, experiential avoidance was by far the most explanatory variable of levels of perceived stress (direct association). Self-compassion was also a significant explanatory variable that was inversely associated with perceived stress, but to a lower degree. Finally, trait mindfulness showed a non-significant additive explanatory power in the regression, but when self-compassion scores did not account for mindfulness and overidentification items, mindfulness was shown as a significant explanatory variable inversely associated with perceived stress. Similarly, experiential avoidance significantly explained the levels 'overload' and 'lack of development' burnout dimensions to a greater degree than self-compassion and mindfulness in our previous study [33]. Thus, both results may be further evidence of a stronger connection between psychopathology and experiential avoidance than between psychopathology and other third-waves constructs (e.g., mindfulness or compassion) in university students.

In our sample, perceived stress was positively associated with experiential avoidance, which could be regarded as a psychological variable with potential risk. In contrast, perceived stress was negatively correlated with self-compassion and mindfulness, which, in turn, could be seen as protective factors. The same scenario was shown regarding the association between experiential avoidance, mindfulness, and self-compassion with burnout dimensions in our previous study [33]. These results can be explained taking the emotion dysregulation model of psychological distress into account [45], which posits that the ways individuals experience and respond to emotions can lead or not to psychological distress. According to this model, motion dysregulation (or misregulation) can occur when the selected emotion regulation strategies do not match the situation or keep the person in an endless struggle to free himself of the unwanted emotions, being experiential avoidance the flagship example among maladaptive strategies. In fact, it is considered a strong transdiagnostic predictor of psychological distress [12]. In contrast, higher mindfulness and self-compassion levels (as a state or trait) are typically related to less psychological distress. For instance, it has been found that low-level mindfulness individuals frequently respond to acute stressors showing a pattern of excessive and blunted responses that could lead to a host of poor health outcomes [16]. In addition, the negative components of self-compassion showed greater associations with psychological distress than the positive counterparts in a recent meta-analytic study [46].

Our findings provide further evidence on the association between experiential avoidance and psychological disturbances [47] and would indicate that stressed students present a tendency to escape or avoid private events (e.g., emotions or memories), regardless of their sociodemographic and academic profile. In this line, previous studies have reported significant associations between experiential avoidance and depression or suicidality in Taiwanese university students [26], as well as between experiential avoidance and burnout syndrome in Spanish undergraduate nursing students [33]. Experiential avoidance would also seem to moderate the positive relationship between anxiety sensitivity and perceived stress in a large community sample of university students from the United States [48]. It is also an important construct in anxiety disorders [49]. In fact, there is a certain amount of anxiety research that demonstrates the mental health costs associated with inflexible development of an avoidance response style, such as the permanence of the experience of bodily arousal in people with panic disorder, being concerned about openly exposing and communicating intense emotional experiences to other people, and the fear of strong emotional impulses in anxiety disorders. The abovementioned mental health costs associated with inflexible psychological features perpetuate anxiety disorders and are related to an experiential avoidance style [49].

Self-compassion was the second-best explanatory variable in explaining perceived stress in our sample. Its significant association with perceived stress would indicate that stressed students tend to be self-critical rather than self-kind against failure and adversity, regardless of their sociodemographic and academic profile. These findings are also consistent with previous studies on university students [50,51], in which self-compassion mediated the negative relationship between perceived stress and anxiety and depression symptoms in a large sample of German university students. Furthermore, it was suggested that self-compassion could reduce anxiety and depression levels [50,51], and may increase students' capacity for managing the emotional demands of their studies [52].

Even though the inverse association between dispositional mindfulness and psychopathological symptoms has been consistently reported [53], our results showed a non-significant relationship between mindfulness as a trait and perceived stress. One possible explanation for that unexpected result was the partial overlap between mindfulness (FFMQ) and self-compassion (SCS) measurements [50]. That is, Neff's model [31] includes mindfulness (vs. overidentification) as one of the three positive components of self-compassion, which is mirrored in the SCS. In fact, when the overlap between these measurements was solved, mindfulness and perceived stress association were found to be statistically significant. Therefore, the non-significant association that was initially found was mostly related to psychometric issues that future research should address using other compassion measurements such as the Sussex-Oxford Compassion Scales [28]. Thus, our results would indicate that stressed students tend to be unaware of the present moment experience and to respond reactively to it. In this line, previous studies have shown the inverse association between mindfulness and perceived stress levels [17,19] as well as the positive association between mindfulness and satisfaction with life [18].

Regarding sociodemographic and academic variables, high total study hours per week, having a partner (vs. being single), being female (vs. being male), and being older were significantly associated with high levels of perceived stress. In addition, perceived family support was found to be a protective variable, a good/very good perception of family support being associated with lower levels of perceived stress. Finally, having children, being in employment, having left the family home, having a scholarship, the academic year, and the number of failed subjects were not significant contributors to higher levels of perceived stress.

As mentioned, higher perceived family support significantly explained lower perceived stress levels in our sample of university students. These findings may provide a basis for explaining how family support may be a relevant buffering variable in the perception of the stress that occurs among undergraduate students [54,55]. For example, in a sample of Chinese university students, family cohesion explained social adjustment by increasing a sense of security [55], altogether psychosocial factors that are known to buffer the harmful effects of stressors on well-being. Conversely, having a partner was positively associated with perceived stress. It is possible that the family is more supportive of difficulties during the university experience than a partner, who may be more of an obstacle or a source of tension during this period.

Regarding the significant positive association found between perceived stress and study hours per week, the perception of overload due to the amount of study hours has previously been associated with stress and also burnout [56,57]. Furthermore, the load of study hours per week constitutes not only an associated perceived stress variable, but also an explanatory factor that could impair academic performance among university students [58]. Finally, as in other studies, being female and of an older age also explained increased perceived stress levels [35,59,60].

Up to now, mindfulness-based interventions for university students have shown promising effectiveness [8,61], though our results suggest that comprehensive programmes more oriented at fostering other crucial process variables for stress reduction such as experiential avoidance

and self-compassion are also needed in students showing greater levels of perceived stress. Therefore, these variables should be considered as targets in programs aimed at fostering well-being and mental health in university students. Programs aimed at increasing certain psychological skills such as Acceptance and Commitment Therapy [62,63] or the Mindful self-compassion program [64] should be considered when choosing interventions for this population. Indeed, it should be noted the importance of the behavioral components of these interventions as an addition to mindfulness. Given that self-compassion and avoidance (or acceptance) are behavioral in nature, this may be a critical addition to mindfulness-based interventions.

Finally, taking into account that our sample was composed of university teaching students it should be noted that psychological flexibility and self-compassion are important aspects of social and emotional competences in teaching professionals' performance. In turn, the social and emotional competence of teachers is a key point to promote healthy student-teacher relationships, effective classroom management, and effective social and emotional learning implementation that contributes to a healthy classroom atmosphere. This healthy classroom atmosphere benefits students' social, emotional and academic learning process which, in turn, has an indirect effect on teachers' social/emotional well-being [65].

### Limitations and future research

These findings should be interpreted with caution regarding the following limitations. Firstly, our findings are based on self-reported measurements and could suffer from social desirability bias. Moreover, two subscales of the mindfulness measurement showed inadequate internal consistency, so only the total score could be used. Secondly, the SCS and AAQ-II have been extensively criticised for being overly saturated with personality traits or distress rather than specifically measuring self-compassion and experiential avoidance, respectively [66–69], a phenomenon that suggests an inflated relationship with perceived stress in the present study. In this regard, several variables that may have had an influence on the relationship between the outcomes were not assessed. For instance, the role of the previous clinical and medical history of participants or personality traits on perceived stress are not computed. Another example is that data was collected at only one point in time, when students have exams and deadlines which may have influenced findings. Fourthly, an *ad hoc* variable was created to evaluate subjective family support, and further studies should also explore this construct with validated questionnaires. Fifthly, results might be biased due to the followed self-selection sampling procedure and the use of two measurement procedures (in-person and online). In that regard, we were not able to check for differences between subjects answering online and via pencil and paper because it was not registered. Sixthly, a deep sample description of university students studying education in their sociodemographic and study-related profile has been depicted, but due to the ethnicity of the participants (all were White), the purposive sample, lack of stratification in the methodological procedure, and the fact that they were only from one university faculty, our results should not be generalisable to the whole population of university students. Finally, participants were aware of the purpose and the general aim of the study, which might have impacted on the results due to social desirability biases.

### Conclusions

In conclusion, mindfulness, self-compassion, and experiential avoidance explained perceived stress levels in university students studying education after controlling for relevant sociodemographic and academic variables. The association found between these variables and perceived stress underlies the need to include them as potential intervention targets, especially in the current post-pandemic times. The higher explanatory power of experiential avoidance in our

analysis demonstrates that although mindfulness-based programmes have been reported as key interventions to promote psychological well-being, other interventions focused on psychological flexibility should also be taken into consideration. Thus, our findings underpin the need for formative interventions that would aim firstly to reduce experiential avoidance levels, secondly to promote self-compassion, and thirdly to use mindfulness training as a psychotherapeutic aid. Adequate efforts should be made by higher education institutions to recognise and address university students' mental health challenges and towards facilitating their mental health recovery, by reducing the long-term consequences of the pandemic on mental health. Thus, these three variables that can be trained and modified could be included as key factors in specific psychological interventions targeted at university students, as previous studies already suggested. Nevertheless, the causal/temporal relationships between experiential avoidance, self-compassion, and mindfulness skills and perceived stress should be explored in future research using more refined study designs.

## Supporting information

**S1 Fig. Normal Probability Plot (P-P) of the regression standardised residual and the scatterplot.**
(TIF)

**S1 Table. Correlations among sociodemographic, academic, psychological variables, and perceived stress.**
(DOCX)

**S2 Table. Hierarchical regression results for perceived stress (complete table).**
(DOCX)

**S3 Table. Hierarchical regression results for perceived stress with the ad hoc SCS-SF (complete SPSS output).**
(DOCX)

## Author Contributions

**Conceptualization:** David Martínez-Rubio, Jesús Montero-Marín, Albert Feliu-Soler.

**Data curation:** David Martínez-Rubio, Jesús Montero-Marín, Albert Feliu-Soler.

**Formal analysis:** Ariadna Colomer-Carbonell, Juan P. Sanabria-Mazo, Cristina Escamilla, Albert Feliu-Soler.

**Investigation:** David Martínez-Rubio, Jesús Montero-Marín, Albert Feliu-Soler.

**Methodology:** Ariadna Colomer-Carbonell, Juan P. Sanabria-Mazo, Albert Feliu-Soler.

**Project administration:** David Martínez-Rubio, Jesús Montero-Marín, Albert Feliu-Soler.

**Resources:** David Martínez-Rubio, Jesús Montero-Marín, Albert Feliu-Soler.

**Supervision:** Jesús Montero-Marín, Juan V. Luciano, Albert Feliu-Soler.

**Validation:** Adrián Pérez-Aranda.

**Visualization:** Adrián Pérez-Aranda.

**Writing – original draft:** Ariadna Colomer-Carbonell, Juan P. Sanabria-Mazo.

**Writing – review & editing:** David Martínez-Rubio, Ariadna Colomer-Carbonell, Juan P. Sanabria-Mazo, Adrián Pérez-Aranda, Jaime Navarrete, Cristina Martínez-Brotóns, Cristina Escamilla, Anna Muro, Jesús Montero-Marín, Juan V. Luciano, Albert Feliu-Soler.

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
