## [Decision Letter · Decision Letter 0]

11 May 2022

PONE-D-21-35822

How Experiential Avoidance, Self-Compassion, and Mindfulness are related to Perceived Stress in a Sample of University Students of Education

PLOS ONE

Dear Dr. Perez-Aranda,

Thank you for submitting your manuscript to PLOS ONE. After careful consideration, we feel that it has merit but does not fully meet PLOS ONE’s publication criteria as it currently stands. Therefore, we invite you to submit a revised version of the manuscript that addresses the points raised during the review process.

The reviewers and I find your research question interesting. However, a number of concerns have been raised, particularly regarding the analysis of the data. The reviewers have provided detailed feedback. Please respond to their points in a revision. 

We look forward to receiving your revised manuscript.

Kind regards,

Natalie J. Shook

Academic Editor

PLOS ONE

3. Peer review at PLOS ONE is not double-blinded (https://journals.plos.org/plosone/s/editorial-and-peer-review-process). For this reason, authors should include in the revised manuscript all the information removed for blind review.

“AC-C has a FI predoctoral contract from AGAUR (FI_B/00216). JPS-M has a PFIS predoctoral contract from the ISCIII (FI20/00034). AP-A has a Sara Borrell postdoctoral contract from the ISCIII (CD20/00181). JVL had a “Miguel Servet” research contract from the ISCIII (CPII19/00003) when the study was conducted. JM-M is supported by the Wellcome Trust Grant (104908/Z/14/Z/). AF-S and JVL acknowledge the funding from the Serra Húnter program (UAB-LE-8015 and UAB-LE-120014, respectively). The funding bodies did not play any role in the analysis and interpretation of data, in the writing of the manuscript, or in the decision to submit the paper for publication.”

“AC-C has a FI predoctoral contract from AGAUR (FI_B/00216). JPS-M has a PFIS predoctoral contract from the ISCIII (FI20/00034). AP-A has a Sara Borrell postdoctoral contract from the ISCIII (CD20/00181). JVL had a “Miguel Servet” research contract from the ISCIII (CPII19/00003) when the study was conducted. JM-M is supported by the Wellcome Trust Grant (104908/Z/14/Z/). AF-S and JVL acknowledge the funding from the Serra Húnter program (UAB-LE-8015 and UAB-LE-120014, respectively). The funding bodies did not play any role in the analysis and interpretation of data, in the writing of the manuscript, or in the decision to submit the paper for publication.”

“AC-C has a FI predoctoral contract from AGAUR (FI_B/00216). JPS-M has a PFIS predoctoral contract from the ISCIII (FI20/00034). AP-A has a Sara Borrell postdoctoral contract from the ISCIII (CD20/00181). JVL had a “Miguel Servet” research contract from the ISCIII (CPII19/00003) when the study was conducted. JM-M is supported by the Wellcome Trust Grant (104908/Z/14/Z/). AF-S and JVL acknowledge the funding from the Serra Húnter program (UAB-LE-8015 and UAB-LE-120014, respectively). The funding bodies did not play any role in the analysis and interpretation of data, in the writing of the manuscript, or in the decision to submit the paper for publication.”

Reviewers' comments:

Reviewer's Responses to Questions

**Comments to the Author**

1. Is the manuscript technically sound, and do the data support the conclusions?

Reviewer #1: Partly

Reviewer #2: Partly

Reviewer #3: Yes

2. Has the statistical analysis been performed appropriately and rigorously? 

Reviewer #1: No

Reviewer #2: No

Reviewer #3: Yes

3. Have the authors made all data underlying the findings in their manuscript fully available?

Reviewer #1: No

Reviewer #2: No

Reviewer #3: No

4. Is the manuscript presented in an intelligible fashion and written in standard English?

Reviewer #1: No

Reviewer #2: Yes

Reviewer #3: Yes

5. Review Comments to the Author

Reviewer #1: I do not agree with the statistical approach used by the authors and have suggested a different analysis (see attachment). The authors stated that the data is available upon request, which does not align with the PLOS data policy. There are grammatical and spelling errors in the manuscript, which I have noted (see attachment).

Reviewer #2: The manuscript reports findings from a cross-sectional survey study of university students in education regarding perceived stress and therapeutically relevant processes. The manuscript is well-written and generally clear. Overarching concerns include: 1) a lack of unifying theme among the processes examined or a framework to unite them therapeutically (which will help with clinical implications) and 2) the potential for bias in the statistical approach that was selected.

Abstract

The authors note that there is a high “prevalence” of perceived stress among college students. Please revise to indicate high levels, as prevalence has another meaning which the authors did not measure.

Introduction

Authors should define the “third wave” and “fourth wave” therapies for readers who may be unfamiliar with what approaches fall within each wave.

Experiential avoidance is defined as the opposite of psychological flexibility, though there is quite a bit of debate about this in the ACT literature. The authors might just define experiential avoidance, mention that this is a process addressed in trying to facilitate psychological flexibility, or note it as a part of psychological inflexibility (which is thought of as the opposite of psychological flexibility in some of the recent literature).

A clearer picture of the processes being examined fit together may be helpful through a unifying theme before each is defined individually. For example, the non-judgment process of mindfulness often includes elements of self-compassion. The act with awareness and nonreactivity facets of mindfulness often include some form of acceptance (or lack of avoidance). A unifying theme would help to illustrate how these are all interrelated processes and justify the later analytic choice.

The aims/objective statement at the end of the introduction could be more clearly phrased. The authors wanted to explore the relative contributions of three therapeutically relevant processes to perceived stress, a critical construct among college students who are studying for a career in a field known to be at risk for burnout, depression, and anxiety. The authors should emphasize their question and its importance here.

Method

Were participants compensated for participating, or did they get course credit?

For the PSQ measure, the authors could provide more information about the original validation study, including construct, convergent, discriminant, or incremental validity. Thank you for noting that your group validated this for use among Spanish university students, and great work – it’s great to see follow-up validation studies being done. The authors later report a category of medium score of stress in the Results – the way this score is derived should be reported in the section regarding this measure.

There is some confusion with the FFMQ-SF scale. You note that it has 20 items with 5 items for 5 subscales, but this results in 25 items. The citation is also for the original scale, not the short form. The authors might also address the low internal consistency of the observing, describing, and non-reactivity subscales. Was the internal consistency for the total score improved?

Regarding the SCS-SF, the authors did not use the subscales as they did for the FFMQ-SF. Can the authors justify this choice? One alternative would be to use total scores for all scales, and if significant associations are uncovered in the regression, conduct an exploratory analysis with the subscales instead of the total score to identify the most important processes.

The authors specify March to May completion dates, but they do not specify the year. Given the implications of the COVID-19 pandemic for university students, specify the year if possible, and if not, whether the data were collected pre- or peri-pandemic.

The authors specify prior studies as justification to recode sociodemographic and academic variables, but they might offer a statistical justification instead. At the very least, as a reviewer, I cannot judge the appropriateness of this approach without breaking the masking of the review process since authors were not specified.

The authors might consider not using a stepwise approach to the regression, as it generally is thought to bias R2 values to be higher, deflate standard errors, and narrow confidence intervals. In addition, this exacerbates collinearity problems, which is likely to be the biggest issue with stepwise and the authors’ data, as the independent variables are at least modestly correlated. The authors should consider hierarchical regression where all variables originally included are reported in the final models. Below are a few citations to consider about stepwise. If the authors choose to keep the stepwise approach, these limitations should be acknowledged thoroughly and justification of this approach should be provided.

Thompson, B. (1995). Stepwise regression and stepwise discriminant analysis need not apply here: A guidelines editorial. Educational and psychological measurement, 55(4), 525-534.

Henderson, D. A., & Denison, D. R. (1989). Stepwise regression in social and psychological research. Psychological Reports, 64(1), 251-257.

Whittingham, M. J., Stephens, P. A., Bradbury, R. B., & Freckleton, R. P. (2006). Why do we still use stepwise modelling in ecology and behaviour?. Journal of animal ecology, 75(5), 1182-1189.

Results

What is the variable “number of failed subjects”? The authors might explain this further in the method, as this is not entirely clear when reported in the results/table.

The interrelations among the processes should be specified in-text as these bear on the potential for multicollinearity in the regression analyses.

I’m not sure how to solve this, as large correlation matrices are hard to manage, but it is very challenging to interpret a correlation matrix that stretches across three pages.

Discussion

The discussion section speaks to the findings and contextualizes them in the broader literature, but this section could benefit from the unifying theme that I suggest in the introduction section comments. This would help to provide readers a clear next step with regard to clinical implications and future research. These processes are all part of Acceptance and Commitment Therapy or more broadly, mindfulness- and acceptance-based therapies, and if discussed as part of these approaches, it will clarify how they all interrelate.

The authors might also provide context around the known risk of educators for burnout, anxiety, depression, and other forms of distress-related disorders. These findings could be use to support prevention efforts during education that will help educators have better occupational health once on the job.

Concerns about the AAQ-II and its modest correlation with negative affect should be acknowledged. This could be the reason for its prediction of perceived stress, and it is a limitation of the measure (not your study).

Tyndall, I., Waldeck, D., Pancani, L., Whelan, R., Roche, B., & Dawson, D. L. (2019). The Acceptance and Action Questionnaire-II (AAQ-II) as a measure of experiential avoidance: Concerns over discriminant validity. Journal of Contextual Behavioral Science, 12, 278-284.

In addition, the method for recruitment and whether compensation was provided should be included. If people self-selected into the study, this should be noted as a limitation.

The authors report change in adjusted R2 and use this as the metric for defining which variables had the most explanatory power. I am curious how much the psychological processes contributed as a group as well, over and above the sociodemographic and academic variables. A hierarchical linear regression without the stepwise approach (with sociodemographic and academic variables in step 1, and the psychological variables in step 2) could answer this question and would important given their interrelations.

Reviewer #3: This paper explores the association between potential protective factors of health and perceived stress in university students. Given the high levels of stress in this group, I believe this paper is timely and needed.

The paper is well written, and analyses are well conducted. My main concern is related to the use of the AAQ-II (Acceptance and Action Questionnaire), which has extensively been reported as overly saturated with personality traits or distress rather than specifically measuring experiential avoidance (Wolgast et al., 2014). I encourage the authors to include this a major limitation in the discussion section and to soften and/or adapt some of the implications they are making in the discussion.

Please find below some additional minor suggestions:

I would recommend that the authors add a paragraph to the introduction explaining the broader rationale and context for 3rd wave interventions in this specific population.

More emphasis needs to be placed on the clinical implications of the findings. The discussion mentions what components should be included when designing an intervention for this group however, the authors do not consider or discuss factors that might not make these interventions feasible. Rather, the authors should consider adjustments that can be made through the university to make uptake of an intervention more likely.

Issues with data collection that need better context:

• Ethnicity of participants not reported – there can be cultural differences influencing certain variables.

• Data was collected on how much family support participants had. How exactly was this measured as it is likely that this was entirely subjective.

• Previous clinical and medical history of participants not reported and nor has this been addressed.

• Authors do not address that data was collected at one time point only, this time point was most likely when students have exams and deadlines which would have influenced findings.

6. PLOS authors have the option to publish the peer review history of their article (what does this mean?). If published, this will include your full peer review and any attached files.

Reviewer #1: No

Reviewer #2: No

Reviewer #3: No

---

## [Author Response · Author response to Decision Letter 0]

23 Jun 2022

Journal requirements

and

Answer: The paper has been formatted according to PLOS ONE’s style requirements.

Answer: We have specified the information about the consent (page 10): “Students were invited to participate at the end of their classes. They were informed about the voluntary nature of participation and the confidentiality of the data. Concretely, it was explained to them that the aim of the study was to investigate the association among their levels of perceived stress, mindfulness, self-compassion, and experiential avoidance. After that, we obtained written informed consent from all students interested in the study. Then, they were asked to answer individually the study measures through a paper-and-pencil survey or via online format (Survey Monkey; 46% of the cases).”

3. Peer review at PLOS ONE is not double-blinded (https://journals.plos.org/plosone/s/editorial-and-peer-review-process). For this reason, authors should include in the revised manuscript all the information removed for blind review.

Answer: Fixed. We have unblinded the manuscript.

“AC-C has a FI predoctoral contract from AGAUR (FI_B/00216). JPS-M has a PFIS predoctoral contract from the ISCIII (FI20/00034). AP-A has a Sara Borrell postdoctoral contract from the ISCIII (CD20/00181). JVL had a “Miguel Servet” research contract from the ISCIII (CPII19/00003) when the study was conducted. JM-M is supported by the Wellcome Trust Grant (104908/Z/14/Z/). AF-S and JVL acknowledge the funding from the Serra Húnter program (UAB-LE-8015 and UAB-LE-120014, respectively). The funding bodies did not play any role in the analysis and interpretation of data, in the writing of the manuscript, or in the decision to submit the paper for publication.”

“AC-C has a FI predoctoral contract from AGAUR (FI_B/00216). JPS-M has a PFIS predoctoral contract from the ISCIII (FI20/00034). AP-A has a Sara Borrell postdoctoral contract from the ISCIII (CD20/00181). JVL had a “Miguel Servet” research contract from the ISCIII (CPII19/00003) when the study was conducted. JM-M is supported by the Wellcome Trust Grant (104908/Z/14/Z/). AF-S and JVL acknowledge the funding from the Serra Húnter program (UAB-LE-8015 and UAB-LE-120014, respectively). The funding bodies did not play any role in the analysis and interpretation of data, in the writing of the manuscript, or in the decision to submit the paper for publication.”

Answer: Funding-related information has been removed from the paper. I confirm that the Funding Statement should read as follows:

AC-C has a FI predoctoral contract from AGAUR (FI_B/00216). JPS-M has a PFIS predoctoral contract from the ISCIII (FI20/00034). AP-A has a Sara Borrell postdoctoral contract from the ISCIII (CD20/00181). JN has a research contract from the Institute of Health Carlos III (ISCIII; ICI20/00080). JM-M has a “Miguel Servet” contract from the ISCIII (CP21/00080). JM-M was supported by the Wellcome Trust Grant (104908/Z/14/Z/) when the study was conducted. AF-S acknowledges the funding from the Serra Húnter program (Generalitat de Catalunya; reference number UAB-LE-8015). The funding bodies did not play any role in the analysis and interpretation of data, in the writing of the manuscript, or in the decision to submit the paper for publication.

“AC-C has a FI predoctoral contract from AGAUR (FI_B/00216). JPS-M has a PFIS predoctoral contract from the ISCIII (FI20/00034). AP-A has a Sara Borrell postdoctoral contract from the ISCIII (CD20/00181). JVL had a “Miguel Servet” research contract from the ISCIII (CPII19/00003) when the study was conducted. JM-M is supported by the Wellcome Trust Grant (104908/Z/14/Z/). AF-S and JVL acknowledge the funding from the Serra Húnter program (UAB-LE-8015 and UAB-LE-120014, respectively). The funding bodies did not play any role in the analysis and interpretation of data, in the writing of the manuscript, or in the decision to submit the paper for publication.”

Answer: I confirm that ‘the funders had no role in study design, data collection and analysis, decision to publish, or preparation of the manuscript’.

Answer: Following your instructions, we have provided a link to OSF in which the data file associated to the present analysis can be downloaded: 

“For transparency and analytical reproducibility purposes, SPSS data can be accessed at OSF: https://osf.io/g2sfp/”

Reviewer #1

Authors: We would like to thank you very much for your kind and thoughtful comments. We will provide an answer after each comment below.

Introduction

Overall, the introduction does not tell a coherent story to work towards setting up the current study aims. There are related ideas spread across several paragraphs that would benefit from being integrated together. Other paragraphs do not seem necessary (e.g., that teachers experience stress), but rather distract from the study aims. Most importantly, the key focus of the study is around how mindfulness, self-compassion, and experiential avoidance relate to stress, but this is lacking within the introduction. I suggest that the authors identify the 1-2 main aims of their paper and reorganize the introduction section accordingly to better help with flow and interpretability.

Answer: Following reviewer’s suggestion, we have entirely rewritten the introduction section following all your considerations. Changes can be checked throughout the whole section. Find below some paragraphs with major changes that address your concerns (pages 2-6): 

“The mental health of university students has become a worldwide concern [1]. It is estimated that around 35% of university students meet the diagnostic criteria for a mental disorder [2]. Furthermore, the pooled prevalence of depression among undergraduates is 25% and 14% for the prevalence of suicide-related outcomes [3]. In fact, their risk of developing a mental illness has significantly increased up to around 50% as a result of the COVID-19 pandemic [4,5]. Overall, university students are particularly vulnerable to psychological distress, which in this population is strongly related to academic underperformance, i.e. failure to fulfil academic obligations, and problematic health behaviours, such as substance use [6,7]. 

Given the high prevalence of mental disorders among university students, there has been a burgeoning interest in detecting potential protective skills that could be enhanced in order to improve their mental health, such as mindfulness, self-compassion, and psychological flexibility [8, 9]. These are three psychological constructs of “third wave” cognitive behavioural therapy, in which the focus is on the relationship between the individual and his/her thoughts and emotions [10].”

[…]

“Many studies have consistently indicated the negative association of mindfulness, compassion, and psychological flexibility with psychological distress in general populations [14,15]. Specifically in university students, for instance, Burger et al. [16] investigated the associations between psychological distress and mindfulness in a sample of 174 university students and found that higher levels of mindfulness (especially the facet of acting with awareness) significantly predicted lower levels of psychological distress.”

[…]

“Regarding compassion, Chan et al. [19] have found in a sample of 536 university students that compassion for others and self-compassion were linked to lower levels of psychological distress.”

[…]

“Regarding psychological flexibility, previous studies have focused on experiential avoidance, which is a process aimed at trying to facilitate psychological flexibility. Experiential avoidance is defined as a “phenomenon that occurs when a person is unwilling to remain in contact with particular private experiences (e.g., bodily sensations, emotions, thoughts, memories, behavioural predispositions) and takes steps to alter the form or frequency of these events and the contexts that occasion them” ([22], p. 1154).”

[…]

“Mindfulness, self-compassion, and experiential avoidance are three interrelated processes. According to Neff [27], there is a reciprocal association between mindfulness and self-compassion, in which each one is necessary for facilitating and enhancing the other.”

[…]

“Finally, previous studies on the association between sociodemographic variables and distress in university students have shown that, for example, gender plays an important role in stress symptomatology since being female is a vulnerability factor for developing it [29,30].”

[…]

“Overall, all these studies conclude that interventions aimed at increasing mindfulness facets, compassion, or psychological flexibility (by reducing experiential avoidance) may help to improve psychological distress in university students (e.g. [16,18,23]). However, very few studies have simultaneously studied the ability of trait mindfulness, compassion, and experiential avoidance, along with sociodemographic and academic variables, to explain levels of psychological distress in university students. Including all those variables in the same model would add specific information about which of these psychological constructs should be more emphasised in mental health interventions for university students.”

[…]

“Therefore, the aim of the current study was to explore the relative contributions of three processes that are therapeutically relevant to perceived stress (i.e. trait mindfulness, self-compassion, and experiential avoidance), a critical construct among university students who are at risk of burnout, depression, and anxiety.”

[…]

Other suggestions:

The authors should include a first, initial paragraph that introduces the topic and the study aims for their paper, before summarizing the literature.

Answer: Following reviewer’s suggestion, we have begun the introduction section speaking about the topic of the study, i.e., the mental health status of university studies and the interest in detecting potential protective skills that could be enhanced to improve their mental health (page 2): 

“The mental health of university students has become a worldwide concern [1]. It is estimated that around 35% of university students meet the diagnostic criteria for a mental disorder [2]. Furthermore, the pooled prevalence of depression among undergraduates is 25% and 14% for the prevalence of suicide-related outcomes [3]. In fact, their risk of developing a mental illness has significantly increased up to around 50% as a result of the COVID-19 pandemic [4,5]. Overall, university students are particularly vulnerable to psychological distress, which in this population is strongly related to academic underperformance, i.e. failure to fulfil academic obligations, and problematic health behaviours, such as substance use [6,7]. 

Given the high prevalence of mental disorders among university students, there has been a burgeoning interest in detecting potential protective skills that could be enhanced in order to improve their mental health, such as mindfulness, self-compassion, and psychological flexibility [8, 9].”

• The focus of the paper is on dispositional mindfulness, but the introduction only briefly mentions how mindfulness trainings are related to stress. The connection between dispositional mindfulness and stress, especially among college students, needs to be described to better set up the paper.

Answer: Following reviewer’s suggestion, we have described the connection between trait mindfulness and stress in university students (page 3): 

“Many studies have consistently indicated the negative association of mindfulness, compassion, and psychological flexibility with psychological distress in general populations [14,15]. Specifically in university students, for instance, Burger et al. [16] investigated the associations between psychological distress and mindfulness in a sample of 174 university students and found that higher levels of mindfulness (especially the facet of acting with awareness) significantly predicted lower levels of psychological distress. Similarly, Muro et al. [17] found that mindfulness was associated with higher levels of life satisfaction in a sample of Spanish university students, a result which is consistent with the multicentre cross-sectional study of Salvarani et al. [18], who found that undergraduate nursing students (n = 622) with higher dispositional mindfulness scores had lower levels of psychological distress.”

• Self-compassion is suggested as a potential mediator of the relationship between mindfulness and happiness. By bringing this up, I wondered if the authors would test this mediation with stress as an outcome. The focus of the paragraph should be about the association between self-compassion and stress among college students.

Answer: Following reviewer’s suggestion, we have focused the paragraph on the association between self-compassion and stress among college students (page 3):

“Regarding compassion, Chan et al. [19] have found in a sample of 536 university students that compassion for others and self-compassion were linked to lower levels of psychological distress. In addition, Rahmandani et al. [20] found a significant negative association between self-compassion and several dimensions of distress (e.g. loss of confidence, anxiety), suggesting that high self-compassion might explain a lower level of distress in university students. Both compassion for others and self-compassion have consistently been found to be related to trait mindfulness (e.g. [21]).”

• Experiential avoidance is discussed as a potential moderator of several associations, including anxiety, addictions, etc. Again, the authors need to discuss literature about how it is related to stress in particular to better set up the study aims.

Answer: Following reviewer’s suggestion, we have described the connection between experiential avoidance and stress in university students (page 3-4): 

“Regarding psychological flexibility, previous studies have focused on experiential avoidance, which is a process aimed at trying to facilitate psychological flexibility. Experiential avoidance is defined as a “phenomenon that occurs when a person is unwilling to remain in contact with particular private experiences (e.g., bodily sensations, emotions, thoughts, memories, behavioural predispositions) and takes steps to alter the form or frequency of these events and the contexts that occasion them” ([22], p. 1154). For instance, Farr et al. [23] found that experiential avoidance was a significant predictor of psychological distress, this positive association being moderated by self-compassion. Specifically, these authors showed that individuals high in self-compassion reported lower levels of depressive symptoms across low to high experiential avoidance levels than individuals with low self-compassion levels [23]. In addition, Chou et al. [24] found in a sample of 500 university students that experiential avoidance was a significant predictor of Internet addiction, significant depression, and suicidality. Finally, experiential avoidance and self-compassion are strongly and negatively associated [25]. Similarly, mindfulness has been found to negatively correlate with experiential avoidance [26].”

• It will also be beneficial to discuss how mindfulness, self-compassion, and experiential avoidance are associated with each other. 

Answer: Following reviewer’s suggestion, we have described the connection between these three variables (pages 3-4): 

“Both compassion for others and self-compassion have consistently been found to be positively related to trait mindfulness (e.g. [21]).”

[…]

“Finally, experiential avoidance and self-compassion are strongly and negatively associated [25]. Similarly, mindfulness has been found to negatively correlate with experiential avoidance [26].”

[…]

“Mindfulness, self-compassion, and experiential avoidance are three interrelated processes. According to Neff [27], there is a reciprocal association between mindfulness and self-compassion, in which each one is necessary for facilitating and enhancing the other. […]. Furthermore, being unjudgmental and acceptance (central facets of mindfulness) can be considered the beneficial counterparts of experiential avoidance [26]. Likewise, mindfulness and experiential avoidance are drawn from very different theoretical contexts, i.e. Eastern philosophy and functional contextualism, respectively.” 

• The study aims should be reframed to focus on exploring associations among mindfulness, self-compassion, and experiential avoidance with stress. Do the authors have any hypotheses?

Answer: Following reviewer’s suggestion, we have reframed the study aims to focus on those associations. The hypotheses have been emphasized (page 5-6): 

“Therefore, the aim of the current study was to explore the relative contributions of three processes that are therapeutically relevant to perceived stress (i.e. trait mindfulness, self-compassion, and experiential avoidance), a critical construct among university students who are at risk of burnout, depression, and anxiety. According to the literature, we expected that mindfulness, self-compassion, and experiential avoidance would explain a significant proportion of the variance of perceived stress after controlling for relevant sociodemographic and academic variables such as age, gender, or having a scholarship. The direction of the relationship between mindfulness/self-compassion and perceived stress was expected to be negative (inverse). Alternatively, a positive relationship was expected between experiential avoidance and stress. No prior hypotheses were established on the strength of the associations due to the scarce number of previous studies.”

Methods

• Additional descriptions of study measures should be included to better interpret some of the measures. How was presence of family support measured (yes/no; good/bad)? How were overall scores created (summed, averaged)?

Answer: Following reviewer’s suggestion, we have included that information. Concretely, we have specified the response options for the sociodemographic and academic variables that were categorical (page 8):

“The following sociodemographic characteristics were recorded using an ad hoc questionnaire (response options): age, gender (male/female), having a partner (yes/no), number of children, perceived presence of family support (insufficient/good/very good), employment (yes/no), and emancipation status (yes/no). Likewise, the following academic variables were recorded: having a scholarship (yes/no), which academic year the student was in, total study hours per week, and number of failed subjects over the previous examination period. Along with this survey, participants completed the aspects described below as part of a pencil-and-paper or online set of measurements.”

Moreover, we have specified how total scores of the questionnaires were calculated (pages 9-10):

“The Perceived Stress Questionnaire-Short Form (PSQ-SF; [36]) […] The total score ranges from 0 to 1 and can be calculated by adding all items, subtracting 24, and dividing by 72 [37]. 

[…]

The Five Facets Mindfulness Questionnaire-Short Form (FFMQ-SF; [38]) […] Two subscales (describing and non-reactivity to inner experience) demonstrated inadequate internal consistency (Cronbach’s α < .70), thus only the total score, which was computed by calculating the sum of all items, was retained for the analyses. Higher scores represent greater levels of mindfulness, with the total scores ranging from 20 to 100 points. 

[…]

The Self-Compassion Scale-Short Form (SCS-SF; [39]) […] A total score can be computed by calculating the sum of all items after reversing those that are negatively worded (scores ranging from 12 to 60). Higher scores indicate more tendency towards self-compassion. 

[…]

The Acceptance and Action Questionnaire-II (AAQ-II; [41]) […] A total score can be calculated by adding up all items, with scores ranging from 7 to 49. Higher scores indicate a higher tendency to present experiential avoidance.”

• Why did the authors choose to explore the five facets of mindfulness instead of an overall index score? This rationale should be included. The non-reactivity subscale demonstrated low reliability, which should be noted as a limitation. Was a Spanish version of the mindfulness scale used? If so, this should be stated.

Answer: We agree with the reviewer. Although we wanted to know the contribution of each of the five facets of mindfulness on perceived stress, not only the non-reactivity subscale, but also the describing subscale showed a low reliability. Thus, we have calculated a total score and used it in the analysis. It showed an adequate reliability. In addition, we have specified that we used the Spanish version of the instrument (page 9):

“The Five Facets Mindfulness Questionnaire-Short Form (FFMQ-SF; [38]) is a 20-item questionnaire that measures the five facets of mindfulness: observing, describing, acting with awareness, non-judging of inner experience, and non-reactivity to inner experience. The response format is a 5-point scale (1 = “never or very rarely true”, 5 = “very often or always true”). Tran et al. [38] validated the German and the Spanish version of this FFMQ-SF. Two subscales (describing and non-reactivity to inner experience) demonstrated inadequate internal consistency (Cronbach’s α < .70), thus only the total score, which was computed by calculating the sum of all items, was retained for the analyses. Higher scores represent greater levels of mindfulness, with the total scores ranging from 20 to 100 points. In the current sample, the total score showed an adequate internal consistency with a Cronbach’s α value of 0.76.”

38. Tran US, Cebolla A, Glück TM, Soler J, Garcia-Campayo J, Moy T von. The Serenity of the Meditating Mind: A Cross-Cultural Psychometric Study on a Two-Factor Higher Order Structure of Mindfulness, Its Effects, and Mechanisms Related to Mental Health among Experienced Meditators. PLOS ONE. 2014;9(10):e110192.

Finally, the inadequate internal consistency of the FFMQ subscales has been mentioned as a limitation of the study (page 20):

“These findings should be interpreted with caution regarding the following limitations. Firstly, our findings are based on self-reported measurements and could suffer from social desirability bias. Moreover, two subscales of the mindfulness measurement showed inadequate internal consistency, so only the total score could be used.”

Procedure

• What were the students informed that the study was about?

Answer: Following reviewer’s suggestion, we have provided that information (page 10): 

“In particular, it was explained to them that the aim of the study was to investigate the association between their levels of perceived stress, mindfulness, self-compassion, and experiential avoidance.”

• What year was the study conducted in?

Answer: Following reviewer’s suggestion, we have provided that information (page 11):

“The survey responses were collected from March 2016 to May 2016.” 

• Were the students compensated for participating?

Answer: Following reviewer’s suggestion, we have provided that information (page 11): 

“There was no payment for participating. Returning a blank survey was accepted without any associated punishment.”

• Were there any differences between students who completed the survey online vs. those who completed the survey using a pencil and paper?

Answer: Regretfully, we are not able to identify which subjects responded online and which ones via paper-and-pencil. This aspect was not registered. Then, we have highlighted it in the limitations section as well as results might have been biased by these two different answering procedures (page 21):

“Fifthly, results might be biased due to the followed self-selection sampling procedure and the use of two measurement procedures (in-person and online). In that regard, we were not able to check for differences between subjects answering online and via pencil and paper because it was not registered.”

Results

• How were missing data handled?

Answer: Missing data were handled with the ‘Exclude cases pairwise’ option. This option excluded the cases only if they are missing the data required for the specific analysis. They will still be included in any of the analyses for which they have the necessary information. It is strongly recommended instead of the ‘exclude cases listwise’ option, which include cases in the analysis only if it has full data on all of the variables included in the analysis. This information has been included in the Data Analysis section (page 12): “Pairwise deletion was the technique used to handle missing data.”

• The authors stated that stress and other variables corresponded to “a medium score range.” What exactly do the authors mean by this? Are there cut-offs for clinically meaningful levels of stress? Are there references for interpreting what these values mean?

Answer: Following reviewers’ suggestions, we have focused on the hierarchical multiple regression. We have removed complementary analyses from the main document, i.e., those that were not necessary to answer the research question. Thus, we have removed this part of the text.

• The paragraph describing the description of the sample repeats all of the info in Table 1. Unless the authors want to describe clinically meaningful cut-off scores based on the means of the variables, I think this information can be removed from the text and only included in the table.

Answer: Following reviewer’s suggestion, we have removed in-text information that was shown in the tables. Now, the paragraph describing the description of the sample provides the following information (page 6): 

“Table 1 displays all the sociodemographic, academic, and psychological characteristics for the total sample. The study sample was composed of 589 university teaching students from the San Vicente Mártir Catholic University of Valencia (Spain), of whom 81.2% were female, and the mean age was 22.13 years old (SD = 3.90; range: 18-48). All participants were White. The inclusion criteria were: (1) being ≥18 years old, (2) being an undergraduate student in the field of education, and (3) being able to understand written Spanish.”

• I suggest that the authors do not use a stepwise regression for their analyses. A stronger approach would be a hierarchical multiple regression based on variables associated with stress at the univariable level (correlations). This would still examine whether mindfulness, self-compassion, and experiential avoidance are associated with stress above the effects of sociodemographic and academic variables.

Answer: We agree with the reviewer. Following reviewer’s suggestion, we have re-analyzed our data conducting a hierarchical multiple regression (enter method, not stepwise) and re-organized the results section according to that. As reviewer suggests, in the hierarchical multiple regression we enter our variables in steps or blocks in a predetermined order (not letting the computer decide, as would be the case for stepwise regression). In the first block, we forced sociodemographic and academic variables responding into the analysis. This has the effect of statistically controlling for these variables. In the second step we entered the other independent variables (mindfulness, self-compassion, and experiential avoidance) into the model as a block being removed the possible effect of sociodemographic and academic variables. This information has been included in pages 11-12:

“A hierarchical multiple regression was carried out to assess the ability of FFMQ-SF (mindfulness), SCS-SF (self-compassion), and AAQ-II (experiential avoidance) measurements to explain PSQ-SF (perceived stress) scores after controlling for the influence of sociodemographic and academic variables. All categorical sociodemographic and academic variables were recoded as dummy variables as follows: gender (0 = male, 1 = female), having a partner (0 = no, 1 = yes), children (0 = no, 1 = yes), perceived presence of family support (0 = no = insufficient, 1 = yes = good/very good), employment (0 = no, 1 = yes), left home (0 = no, 1 = yes), having a scholarship (0 = no, 1 = yes), which academic year the student was in (each academic year recoded as dummy variable; 0 = no, 1 = yes). These categorical variables along with age, total study hours per week, and number of failed subjects over the previous examination period were entered at Step 1. Additionally, FFMQ-SF, SCS-SF, and AAQ-II scores were entered at Step 2. Pairwise deletion was the technique used to handle missing data. Semi-partial correlation coefficients (sr) were examined to get an indication of the unique contribution of each variable to the total R square. Specifically, if this coefficient is squared, the percentage of unique variance explained by an independent variable in the dependent variable is obtained [44]. Preliminary analyses were conducted to check the assumptions of multicollinearity, outliers, normality, linearity, and homoscedasticity (S1 and S2 Tables and S3 Fig.), with no major violations noted. Continuing along the path laid down by previous studies [45], the hierarchical multiple regression was re-tested including an ad hoc SCS-SF total score for which the mindfulness and overidentification scores were removed (S4 Table). This analysis was performed due to the partial overlap between trait mindfulness and self-compassion measurements (i.e. FFMQ-SF and SCS-SF, respectively; [28]).”

Tables

Table 1

• The authors should include a note under the table explaining what the abbreviations for the measures correspond to.

• Please add the range of scores for each measure within your sample.

Answer: Fixed. We have explained the abbreviations in a foot note under the table. Following reviewer’s suggestion, we have included also the possible range of the scores. Please, see below the last part of Table 1 (page 8):

“Table 1. Sociodemographic, Academic, and Psychological Characteristics of the Sample (n = 589)

Psychological variables M (SD) Minimum Maximum

PSQ-SF (0 - 1) 0.42 (0.17) 0.01 0.97

FFMQ-SF (20 - 100) 63.70 (9.52) 25 97

SCS-SF (12 - 60) 36.74 (8.51) 12 60

AAQ-II (7 - 49) 22.85 (9.11) 7 48

Note. n = frequencies; % = percentages; M = mean; SD = standard deviation; PSQ-SF = Perceived Stress Questionnaire-Short Form; FFMQ-SF = Five Facets Mindfulness Questionnaire-Short Form; SCS-SF = Self-Compassion Scale-Short Form; AAQ-II = The Acceptance and Action Questionnaire-II. Range of possible scores are shown in brackets.”

Table 2

• This table is very large and difficult to read.

• The key study variables should be listed at the top of the table (stress, mindfulness, self-compassion, and experiential avoidance).

• The authors should include a note under the table explaining how variables were dichotomized. For example, what does a negative vs. positive correlation for gender mean?

Answer: We agree with the reviewer in the inconvenience of the table size. Moreover, after reflecting about its importance in the study, we have decided to reduced it and include it as part of the Supplementary Materials (S1 Table). It shows complementary results. Furthermore, for more clarity, we have improved its readability following reviewer’s suggestions (options for dichotomous variables and a note explaining the abbreviations were included). Finally, how variables were dichotomized is included in the Data Analysis section (page 11):

“All categorical sociodemographic and academic variables were recoded as dummy variables as follows: gender (0 = male, 1 = female), having a partner (0 = no, 1 = yes), children (0 = no, 1 = yes), perceived presence of family support (0 = no = insufficient, 1 = yes = good/very good), employment (0 = no, 1 = yes), left home (0 = no, 1 = yes), having a scholarship (0 = no, 1 = yes), which academic year the student was in (each academic year recoded as dummy variable; 0 = no, 1 = yes).”

Discussion

• The first paragraph does not highlight the main findings from the regression.

Answer: Following reviewer’s suggestion, we have rewritten the first paragraph of the discussion section. Now it highlights the main findings from the regression (pages 16-17): 

“This study examined the explanatory power of mindfulness, self-compassion, and experiential avoidance with regards to perceived stress in a sample of university students while controlling for the influence of relevant sociodemographic and academic variables. When all of them were taken into consideration as potential explanatory variables of perceived stress in the regression model, mindfulness, self-compassion, and experiential avoidance explained a greater variance of perceived stress than all sociodemographic and academic variables together. In particular, experiential avoidance was by far the most explanatory variable of levels of perceived stress (direct association). Self-compassion was also a significant explanatory variable that was inversely associated with perceived stress, but to a lower degree. Finally, trait mindfulness showed marginal additive explanatory power in the regression, but when self-compassion scores did not account for mindfulness and overidentification items, mindfulness was shown as a significant explanatory variable inversely associated with perceived stress. Regarding sociodemographic and academic variables, high total study hours per week, having a partner (vs. being single), being female (vs. being male), and being older were significantly associated with high levels of perceived stress. In addition, perceived family support was found to be a protective variable, a good/very good perception of family support being associated with lower levels of perceived stress. Finally, having children, being in employment, having left the family home, having a scholarship, the academic year, and the number of failed subjects were not significant contributors to higher levels of perceived stress. Thus, perceived stress in our sample was positively associated with experiential avoidance, which could be regarded as a psychological variable with potential risk. In contrast, perceived stress was negatively correlated with self-compassion and mindfulness, which, in turn, could be seen as protective factors.”

• Some facets of mindfulness were not associated in the direction as other facets. Why do the authors think that is?

• Why do the authors think that the acting with awareness facet in particular was important, relative to the other facets? What are the implications of these findings? 

Answer: Finally, we only used the total score of the FFMQ-SF. Then, information regarding the facets was removed. It has been explained in the Instruments section (page 9):

“Two subscales (describing and non-reactivity to inner experience) demonstrated inadequate internal consistency (Cronbach’s α < .70), thus only the total score, which was computed by calculating the sum of all items, was retained for the analyses.”

Also, we acknowledged the lack of reliability of the FFMQ-SF facets as a limitation of the study (page 20):

“Moreover, two subscales of the mindfulness measurement showed inadequate internal consistency, so only the total score could be used.”

• The authors mention how self-compassion overlaps with mindfulness. Indeed, self-compassion includes a component of mindfulness. What are the potential methodological concerns with using both constructs in the regression model? Did the authors test self-compassion with the mindfulness component removed?

Answer: The overlap between variables was not a methodological concern for our analysis, multicollinearity analysis (correlations among variables, tolerance and VIF coefficients shown in supplementary material) demonstrated significant but not large enough correlations among these independent variables to preclude regression as well as admissible tolerance and VIF values. The origin of the overlap relies on the theoretical base of the self-compassion measure. However, following reviewer’s suggestion we re-tested the multiple regression including self-compassion with the mindfulness and overidentification components removed. The complete output was included as Supplementary Material (S4 table) and results were commented in the main document. This procedure was explained in data analysis (page 12):

“Continuing along the path laid down by previous studies [45], the hierarchical multiple regression was re-tested including an ad hoc SCS-SF total score for which the mindfulness and overidentification scores were removed (S4 Table). This analysis was performed due to the partial overlap between trait mindfulness and self-compassion measurements (i.e. FFMQ-SF and SCS-SF, respectively; [28]).”

Here, the respective paragraph in the results section (page 16):

 “When the hierarchical multiple regression was re-tested with the SCS-SF total score calculated without adding the mindfulness and overidentification items scores (S4 Table), FFMQ-SF did significantly explain PSQ-SF scores (sr = -0.07; p = .015), along with the aforementioned variables. In particular, AAQ-II, SCS-SF, and FFMQ-SF scores uniquely explained around 11%, 2%, and 0.55%, respectively, of the variance in PSQ-SF scores.”

Finally, the paragraph in the discussion in which we explain this result (pages 18-19):

“Even though the inverse association between dispositional mindfulness and psychopathological symptoms has been consistently reported [52], our results showed a marginally significant relationship between mindfulness as a trait and perceived stress. One possible explanation for that unexpected result was the partial overlap between mindfulness (FFMQ) and self-compassion (SCS) measurements [49]. That is, Neff’s model [27] includes mindfulness (vs. overidentification) as one of the three positive components of self-compassion, which is mirrored in the SCS. In fact, when the overlap between these measurements was solved, mindfulness and perceived stress association were found to be statistically significant. Therefore, the marginal association that was initially found was mostly related to psychometric issues that future research should address using other compassion measurements such as the Sussex-Oxford Compassion Scales [21].”

• Having an in-person, pencil and paper survey could cause potential bias in responses. This should be discussed.

Answer: Following reviewer’s suggestion, the data recollection issue has been included as a limitation of the study (pages 21):

“Fifthly, results might be biased due to the followed self-selection sampling procedure and the use of two measurement procedures (in-person and online). In that regard, we were not able to check for differences between subjects answering online and via pencil and paper because it was not registered.”

Other

• Sometimes abbreviations for stress “PSQ” are used and other times it is referred to as “stress”. Be more consistent throughout the text to reduce confusion.

• Sometimes the variables are referred to as “academic” variables and other time “occupational” variables (table titles; section headings). Be more consistent throughout the text to reduce confusion.

Answer: Following reviewer’s suggestion, we have used more consistently stress abbreviations throughout the paper and the term academic variables has been used, avoiding variants and synonyms. Concretely, we referred to stress as PSQ in the data analysis and results sections, in the rest of the document we said perceived stress. 

• There are several grammatical and spelling errors.

Answer: Following reviewer’s suggestion, the revised version of the manuscript has been proofread by a native English speaker with experience in science writing.

• Have the data from this study been reported elsewhere?

Answer: Data from this study has been uploaded in OSF: https://osf.io/g2sfp/.

Reviewer #2

The manuscript reports findings from a cross-sectional survey study of university students in education regarding perceived stress and therapeutically relevant processes. The manuscript is well-written and generally clear. Overarching concerns include: 1) a lack of unifying theme among the processes examined or a framework to unite them therapeutically (which will help with clinical implications) and 2) the potential for bias in the statistical approach that was selected.

Authors: We would like to thank you very much for your kind and thoughtful comments. We will provide an answer after each comment below.

Abstract

The authors note that there is a high “prevalence” of perceived stress among college students. Please revise to indicate high levels, as prevalence has another meaning which the authors did not measure.

Answer: Following reviewer’s suggestion, we have modified that sentence (page 1): 

“University students constitute a population that is highly vulnerable to developing mental health problems, such as distress.”

Introduction

Authors should define the “third wave” and “fourth wave” therapies for readers who may be unfamiliar with what approaches fall within each wave.

Answer: Following reviewer’s suggestion, we have briefly explained the core feature of third wave therapies. Also, we have removed the mention to fourth wave therapies in order to simplify the introduction section (page 2): 

“Given the high prevalence of mental disorders among university students, there has been a burgeoning interest in detecting potential protective skills that could be enhanced in order to improve their mental health, such as mindfulness, self-compassion, and psychological flexibility [8, 9]. These are three psychological constructs of “third wave” cognitive behavioural therapy, in which the focus is on the relationship between the individual and his/her thoughts and emotions [10].”

Experiential avoidance is defined as the opposite of psychological flexibility, though there is quite a bit of debate about this in the ACT literature. The authors might just define experiential avoidance, mention that this is a process addressed in trying to facilitate psychological flexibility, or note it as a part of psychological inflexibility (which is thought of as the opposite of psychological flexibility in some of the recent literature).

Answer: Following reviewer’s suggestion, we have added the original definition of experiential avoidance and included the reviewer’s idea (page 3):

“Regarding psychological flexibility, previous studies have focused on experiential avoidance, which is a process aimed at trying to facilitate psychological flexibility. Experiential avoidance is defined as a “phenomenon that occurs when a person is unwilling to remain in contact with particular private experiences (e.g., bodily sensations, emotions, thoughts, memories, behavioural predispositions) and takes steps to alter the form or frequency of these events and the contexts that occasion them” ([22], p. 1154).”

A clearer picture of the processes being examined fit together may be helpful through a unifying theme before each is defined individually. For example, the non-judgment process of mindfulness often includes elements of self-compassion. The act with awareness and nonreactivity facets of mindfulness often include some form of acceptance (or lack of avoidance). A unifying theme would help to illustrate how these are all interrelated processes and justify the later analytic choice.

Answer: Following reviewer’s suggestion, we have added a paragraph in which we describe how these processes interrelate (pages 3-4):

“Both compassion for others and self-compassion have consistently been found to be positively related to trait mindfulness (e.g. [21]). […] Finally, experiential avoidance and self-compassion are strongly and negatively associated [25]. Similarly, mindfulness has been found to negatively correlate with experiential avoidance [26].

Mindfulness, self-compassion, and experiential avoidance are three interrelated processes. According to Neff [27], there is a reciprocal association between mindfulness and self-compassion, in which each one is necessary for facilitating and enhancing the other. For example, a self-compassionate response requires a mindful approach to painful thoughts and feelings. However, an important distinction between them is that mindfulness is applied to pleasant, neutral, or unpleasant experiences, whereas self-compassion is focused on suffering. Moreover, self-compassion is focused on the global self, meanwhile mindfulness can be applied to the self as well as to thoughts, feelings, and sensations [28]. Furthermore, being unjudgmental and acceptance (central facets of mindfulness) can be considered the beneficial counterparts of experiential avoidance [26]. Likewise, mindfulness and experiential avoidance are drawn from very different theoretical contexts, i.e. Eastern philosophy and functional contextualism, respectively.” 

The aims/objective statement at the end of the introduction could be more clearly phrased. The authors wanted to explore the relative contributions of three therapeutically relevant processes to perceived stress, a critical construct among college students who are studying for a career in a field known to be at risk for burnout, depression, and anxiety. The authors should emphasize their question and its importance here.

Answer: Following reviewer’s suggestions, we have rephrased the objective statement and modified the whole paragraph (pages 5-6):

“Therefore, the aim of the current study was to explore the relative contributions of three processes that are therapeutically relevant to perceived stress (i.e. trait mindfulness, self-compassion, and experiential avoidance), a critical construct among university students who are at risk of burnout, depression, and anxiety. According to the literature, we expected that mindfulness, self-compassion, and experiential avoidance would explain a significant proportion of the variance of perceived stress after controlling for relevant sociodemographic and academic variables such as age, gender, or having a scholarship. The direction of the relationship between mindfulness/self-compassion and perceived stress was expected to be negative (inverse). Alternatively, a positive relationship was expected between experiential avoidance and stress. No prior hypotheses were established on the strength of the associations due to the scarce number of previous studies.”

Method

Were participants compensated for participating, or did they get course credit?

Answer: Following reviewer’s suggestion, we have specified that participants were not compensated for participating. Likewise, as mentioned, no participation was accepted without any associated punishment (page 11):

“The survey response time was approximately 20 minutes. There was no payment for participating. Returning a blank survey was accepted without any associated punishment.”

For the PSQ measure, the authors could provide more information about the original validation study, including construct, convergent, discriminant, or incremental validity. Thank you for noting that your group validated this for use among Spanish university students, and great work – it’s great to see follow-up validation studies being done. The authors later report a category of medium score of stress in the Results – the way this score is derived should be reported in the section regarding this measure.

Answer: Following reviewer’s suggestion, we have improved the description of the PSQ as follows. Also, we have removed that part of the results section. Following previous reviewers’ suggestions, we removed complementary analysis and focused on the main analysis of the paper, i.e., the hierarchical multiple regression, which answers the research question (page 9):

“The Perceived Stress Questionnaire-Short Form (PSQ-SF; [36]) is a 24-item scale that assesses perceived stress during the previous 30 days. The response format is a 4-point scale (from 1 = “almost never” to 4 = “almost always”). This questionnaire was validated for Spanish university students and showed adequate internal consistency and construct validity [37]. The total score ranges from 0 to 1 and can be calculated by adding all items, subtracting 24, and dividing by 72 [37]. Higher scores represent greater levels of perceived stress. The PSQ has demonstrated excellent internal consistency in the present sample with a Cronbach’s α value of 0.93.”

There is some confusion with the FFMQ-SF scale. You note that it has 20 items with 5 items for 5 subscales, but this results in 25 items. The citation is also for the original scale, not the short form. The authors might also address the low internal consistency of the observing, describing, and non-reactivity subscales. Was the internal consistency for the total score improved?

Answer: Following reviewer’s suggestion, we have improved the description of this measure and addressed the problem of internal consistency of the FFMQ subscales. As you can see below, only the total score was calculated because the poor reliability of some facets (page 9):

“The Five Facets Mindfulness Questionnaire-Short Form (FFMQ-SF; [38]) is a 20-item questionnaire that measures the five facets of mindfulness: observing, describing, acting with awareness, non-judging of inner experience, and non-reactivity to inner experience. The response format is a 5-point scale (1 = “never or very rarely true”, 5 = “very often or always true”). Tran et al. [38] validated the German and the Spanish version of this FFMQ-SF. Two subscales (describing and non-reactivity to inner experience) demonstrated inadequate internal consistency (Cronbach’s α < .70), thus only the total score, which was computed by calculating the sum of all items, was retained for the analyses. Higher scores represent greater levels of mindfulness, with the total scores ranging from 20 to 100 points. In the current sample, the total score showed an adequate internal consistency with a Cronbach’s α value of 0.76.”

Regarding the SCS-SF, the authors did not use the subscales as they did for the FFMQ-SF. Can the authors justify this choice? One alternative would be to use total scores for all scales, and if significant associations are uncovered in the regression, conduct an exploratory analysis with the subscales instead of the total score to identify the most important processes.

Answer: Following reviewer’s suggestion, we have used the total scores of all measures. You can find below which variables were included in the regression (page 11): 

“A hierarchical multiple regression was carried out to assess the ability of FFMQ-SF (mindfulness), SCS-SF (self-compassion), and AAQ-II (experiential avoidance) measurements to explain PSQ-SF (perceived stress) scores after controlling for the influence of sociodemographic and academic variables.”

The authors specify March to May completion dates, but they do not specify the year. Given the implications of the COVID-19 pandemic for university students, specify the year if possible, and if not, whether the data were collected pre- or peri-pandemic.

Answer: Following reviewer’s suggestion, we have included this information (page 11):

“The survey responses were collected from March 2016 to May 2016.”

The authors specify prior studies as justification to recode sociodemographic and academic variables, but they might offer a statistical justification instead. At the very least, as a reviewer, I cannot judge the appropriateness of this approach without breaking the masking of the review process since authors were not specified.

Answer: Following editor’s and reviewer’s suggestion, we have unblinded the manuscript, so now all references can be checked. The recodification of categorial sociodemographic and academic variables have been better explained (page 11):

“All categorical sociodemographic and academic variables were recoded as dummy variables as follows: gender (0 = male, 1 = female), having a partner (0 = no, 1 = yes), children (0 = no, 1 = yes), perceived presence of family support (0 = no = insufficient, 1 = yes = good/very good), employment (0 = no, 1 = yes), left home (0 = no, 1 = yes), having a scholarship (0 = no, 1 = yes), which academic year the student was in (each academic year recoded as dummy variable; 0 = no, 1 = yes).”

The authors might consider not using a stepwise approach to the regression, as it generally is thought to bias R2 values to be higher, deflate standard errors, and narrow confidence intervals. In addition, this exacerbates collinearity problems, which is likely to be the biggest issue with stepwise and the authors’ data, as the independent variables are at least modestly correlated. The authors should consider hierarchical regression where all variables originally included are reported in the final models. Below are a few citations to consider about stepwise. If the authors choose to keep the stepwise approach, these limitations should be acknowledged thoroughly and justification of this approach should be provided.

Thompson, B. (1995). Stepwise regression and stepwise discriminant analysis need not apply here: A guidelines editorial. Educational and psychological measurement, 55(4), 525-534.

Henderson, D. A., & Denison, D. R. (1989). Stepwise regression in social and psychological research. Psychological Reports, 64(1), 251-257.

Whittingham, M. J., Stephens, P. A., Bradbury, R. B., & Freckleton, R. P. (2006). Why do we still use stepwise modelling in ecology and behaviour?. Journal of animal ecology, 75(5), 1182-1189.

Answer: We agree with the reviewer’s comment, so we have computed a hierarchical multiple regression analysis (pages 11-12):

“A hierarchical multiple regression was carried out to assess the ability of FFMQ-SF (mindfulness), SCS-SF (self-compassion), and AAQ-II (experiential avoidance) measurements to explain PSQ-SF (perceived stress) scores after controlling for the influence of sociodemographic and academic variables. All categorical sociodemographic and academic variables were recoded as dummy variables as follows: gender (0 = male, 1 = female), having a partner (0 = no, 1 = yes), children (0 = no, 1 = yes), perceived presence of family support (0 = no = insufficient, 1 = yes = good/very good), employment (0 = no, 1 = yes), left home (0 = no, 1 = yes), having a scholarship (0 = no, 1 = yes), which academic year the student was in (each academic year recoded as dummy variable; 0 = no, 1 = yes). These categorical variables along with age, total study hours per week, and number of failed subjects over the previous examination period were entered at Step 1. Additionally, FFMQ-SF, SCS-SF, and AAQ-II scores were entered at Step 2. Pairwise deletion was the technique used to handle missing data. Semi-partial correlation coefficients (sr) were examined to get an indication of the unique contribution of each variable to the total R square. Specifically, if this coefficient is squared, the percentage of unique variance explained by an independent variable in the dependent variable is obtained [44]. Preliminary analyses were conducted to check the assumptions of multicollinearity, outliers, normality, linearity, and homoscedasticity (S1 and S2 Tables and S3 Fig.), with no major violations noted. Continuing along the path laid down by previous studies [45], the hierarchical multiple regression was re-tested including an ad hoc SCS-SF total score for which the mindfulness and overidentification scores were removed (S4 Table). This analysis was performed due to the partial overlap between trait mindfulness and self-compassion measurements (i.e. FFMQ-SF and SCS-SF, respectively; [28]).”

Results

What is the variable “number of failed subjects”? The authors might explain this further in the method, as this is not entirely clear when reported in the results/table.

Answer: The variable “number of failed subjects” means the number of failed subjects over the previous examination period. Following reviewer’s suggestion, the description of the sociodemographic and academic variables has been updated as follows (page 8):

“The following sociodemographic characteristics were recorded using an ad hoc questionnaire (response options): age, gender (male/female), having a partner (yes/no), number of children, perceived presence of family support (insufficient/good/very good), employment (yes/no), and emancipation status (yes/no). Likewise, the following academic variables were recorded: having a scholarship (yes/no), which academic year the student was in, total study hours per week, and number of failed subjects over the previous examination period. Along with this survey, participants completed the aspects described below as part of a pencil-and-paper or online set of measurements.”

The interrelations among the processes should be specified in-text as these bear on the potential for multicollinearity in the regression analyses. I’m not sure how to solve this, as large correlation matrices are hard to manage, but it is very challenging to interpret a correlation matrix that stretches across three pages.

Answer: Following reviewers’ considerations, we have decided to include this table as supplementary material. As well, it has been simplified. Regarding multicollinearity and other multiple regression assumptions, information provided as supplementary material shows tolerable correlations, tolerance, and VIF values (page 12):

“Preliminary analyses were conducted to check the assumptions of multicollinearity, outliers, normality, linearity, and homoscedasticity (S1 and S2 Tables and S3 Fig.), with no major violations noted.”

Discussion

The discussion section speaks to the findings and contextualizes them in the broader literature, but this section could benefit from the unifying theme that I suggest in the introduction section comments. This would help to provide readers a clear next step with regard to clinical implications and future research. These processes are all part of Acceptance and Commitment Therapy or more broadly, mindfulness- and acceptance-based therapies, and if discussed as part of these approaches, it will clarify how they all interrelate. The authors might also provide context around the known risk of educators for burnout, anxiety, depression, and other forms of distress-related disorders. These findings could be use to support prevention efforts during education that will help educators have better occupational health once on the job.

Answer: Following reviewer’s suggestion, we have emphasized the potential usefulness of our results to inform intervention in university students. These important issues have been discussed in the following manner: 

(Page 17) “Thus, perceived stress in our sample was positively associated with experiential avoidance, which could be regarded as a psychological variable with potential risk. In contrast, perceived stress was negatively correlated with self-compassion and mindfulness, which, in turn, could be seen as protective factors.” 

(Page 19) “Up to now, mindfulness-based interventions for university students have shown promising effectiveness [8,57], though our results suggest that comprehensive programmes more oriented at fostering other crucial process variables for stress reduction such as experiential avoidance and self-compassion are also needed in students showing greater levels of perceived stress. Therefore, these variables should be considered as targets in programmes aimed at fostering wellbeing and mental health in university students. Programmes aimed at increasing certain psychological skills such as Acceptance and Commitment Therapy (ACT; [58,59]) or the Mindful self-compassion programme [60] should be considered when choosing interventions for this population.”

(Pages 21-22): “Thus, our findings underpin the need for formative interventions that would aim firstly to reduce experiential avoidance levels, secondly to promote self-compassion, and thirdly to use mindfulness training as a psychotherapeutic aid. Adequate efforts should be made by higher education institutions to recognise and address university students’ mental health challenges and towards facilitating their mental health recovery, by reducing the long-term consequences of the pandemic on mental health. Thus, these three variables that can be trained and modified could be included as key factors in specific psychological interventions targeted at university students, as previous studies already suggested.”

Finally, please, excuse our modesty when informing about the clinical and practical implications of our study. In fact, it is our view that no single study can be used to inform clinical, policy, prevention, or practice implications. Each study provides suggestive evidence, but whether the findings of the study can be replicated is questionable, especially given the limitations of the study. Thus, the findings of no one study should be used to tell readers what they ought to be doing based on that study. The point is, how can one study with methodological limitations provide directions for clinical work or practice when there is a whole branch of psychology dealing with evidence-based research to inform treatments?

Concerns about the AAQ-II and its modest correlation with negative affect should be acknowledged. This could be the reason for its prediction of perceived stress, and it is a limitation of the measure (not your study).

Tyndall, I., Waldeck, D., Pancani, L., Whelan, R., Roche, B., & Dawson, D. L. (2019). The Acceptance and Action Questionnaire-II (AAQ-II) as a measure of experiential avoidance: Concerns over discriminant validity. Journal of Contextual Behavioral Science, 12, 278-284.

Answer: Following reviewer’s suggestion we have included that aspect about the AAQ-II as a limitation of the study (page 20): “These findings should be interpreted with caution regarding the following limitations. Firstly, our findings are based on self-reported measurements and could suffer from social desirability bias. Moreover, two subscales of the mindfulness measurement showed inadequate internal consistency, so only the total score could be used. Secondly, the SCS and AAQ-II have been extensively criticised for being overly saturated with personality traits or distress rather than specifically measuring self-compassion and experiential avoidance, respectively [62-65], a phenomenon that suggests an inflated relationship with perceived stress in the present study.”

In addition, the method for recruitment and whether compensation was provided should be included. If people self-selected into the study, this should be noted as a limitation.

Answer: Following reviewer’s suggestion, we have included that information in the methods section (page 11): “The survey response time was approximately 20 minutes. There was no payment for participating. Returning a blank survey was accepted without any associated punishment.”; also in the limitations section (page 21): “Fifthly, results might be biased due to the followed self-selection sampling procedure and the use of two measurement procedures (in-person and online).”

The authors report change in adjusted R2 and use this as the metric for defining which variables had the most explanatory power. I am curious how much the psychological processes contributed as a group as well, over and above the sociodemographic and academic variables. A hierarchical linear regression without the stepwise approach (with sociodemographic and academic variables in step 1, and the psychological variables in step 2) could answer this question and would important given their interrelations.

Answer: Following reviewer’s suggestion we have computed a hierarchical multiple regression analysis. In addition to R square, we interpreted semipartial correlation coefficients, which give us information about the explanatory contribution of each variable when the others in the model are controlled for (pages 11-12):

“A hierarchical multiple regression was carried out to assess the ability of FFMQ-SF (mindfulness), SCS-SF (self-compassion), and AAQ-II (experiential avoidance) measurements to explain PSQ-SF (perceived stress) scores after controlling for the influence of sociodemographic and academic variables. All categorical sociodemographic and academic variables were recoded as dummy variables as follows: gender (0 = male, 1 = female), having a partner (0 = no, 1 = yes), children (0 = no, 1 = yes), perceived presence of family support (0 = no = insufficient, 1 = yes = good/very good), employment (0 = no, 1 = yes), left home (0 = no, 1 = yes), having a scholarship (0 = no, 1 = yes), which academic year the student was in (each academic year recoded as dummy variable; 0 = no, 1 = yes). These categorical variables along with age, total study hours per week, and number of failed subjects over the previous examination period were entered at Step 1. Additionally, FFMQ-SF, SCS-SF, and AAQ-II scores were entered at Step 2. Pairwise deletion was the technique used to handle missing data. Semi-partial correlation coefficients (sr) were examined to get an indication of the unique contribution of each variable to the total R square. Specifically, if this coefficient is squared, the percentage of unique variance explained by an independent variable in the dependent variable is obtained [44]. Preliminary analyses were conducted to check the assumptions of multicollinearity, outliers, normality, linearity, and homoscedasticity (S1 and S2 Tables and S3 Fig.), with no major violations noted. Continuing along the path laid down by previous studies [45], the hierarchical multiple regression was re-tested including an ad hoc SCS-SF total score for which the mindfulness and overidentification scores were removed (S4 Table). This analysis was performed due to the partial overlap between trait mindfulness and self-compassion measurements (i.e. FFMQ-SF and SCS-SF, respectively; [28]).”

Reviewer #3

Authors: We would like to thank you very much for your kind and thoughtful comments. We will provide an answer after each comment below.

This paper explores the association between potential protective factors of health and perceived stress in university students. Given the high levels of stress in this group, I believe this paper is timely and needed. The paper is well written, and analyses are well conducted. My main concern is related to the use of the AAQ-II (Acceptance and Action Questionnaire), which has extensively been reported as overly saturated with personality traits or distress rather than specifically measuring experiential avoidance (Wolgast et al., 2014). I encourage the authors to include this a major limitation in the discussion section and to soften and/or adapt some of the implications they are making in the discussion.

Answer: Following reviewer’s suggestion we have included that aspect about the AAQ-II as a limitation of the study (page 20): “Secondly, the SCS and AAQ-II have been extensively criticised for being overly saturated with personality traits or distress rather than specifically measuring self-compassion and experiential avoidance, respectively [62-65], a phenomenon that suggests an inflated relationship with perceived stress in the present study.”

Please find below some additional minor suggestions:

I would recommend that the authors add a paragraph to the introduction explaining the broader rationale and context for 3rd wave interventions in this specific population.

Answer: Following reviewer’s suggestion we have emphasized the importance of mindfulness, compassion, and psychological flexibility constructs in the studied population. See below the new text (pages 2-4):

“Given the high prevalence of mental disorders among university students, there has been a burgeoning interest in detecting potential protective skills that could be enhanced in order to improve their mental health, such as mindfulness, self-compassion, and psychological flexibility [8, 9]. These are three psychological constructs of “third wave” cognitive behavioural therapy, in which the focus is on the relationship between the individual and his/her thoughts and emotions [10]. Firstly, mindfulness can be defined as a two-component mood that involves the self-regulation of attention, i.e. paying attention to the immediate experience (including internal experiences such as thoughts), and a curious, open, and accepting orientation towards it [11]. Secondly, compassion is a cognitive, affective, and behavioural process that involves recognising suffering, common humanity, empathy, tolerance for uncomfortable feelings, and motivation to act or acting to alleviate the suffering of others and/or oneself [12]. Finally, psychological flexibility is defined as the ability to consciously make contact with the present moment and behave in a way that serves valued goals [13]. 

Many studies have consistently indicated the negative association of mindfulness, compassion, and psychological flexibility with psychological distress in general populations [14,15]. Specifically in university students, for instance, Burger et al. [16] investigated the associations between psychological distress and mindfulness in a sample of 174 university students and found that higher levels of mindfulness (especially the facet of acting with awareness) significantly predicted lower levels of psychological distress. Similarly, Muro et al. [17] found that mindfulness was associated with higher levels of life satisfaction in a sample of Spanish university students, a result which is consistent with the multicentre cross-sectional study of Salvarani et al. [18], who found that undergraduate nursing students (n = 622) with higher dispositional mindfulness scores had lower levels of psychological distress. 

Regarding compassion, Chan et al. [19] have found in a sample of 536 university students that compassion for others and self-compassion were linked to lower levels of psychological distress. In addition, Rahmandani et al. [20] found a significant negative association between self-compassion and several dimensions of distress (e.g. loss of confidence, anxiety), suggesting that high self-compassion might explain a lower level of distress in university students. Both compassion for others and self-compassion have consistently been found to be positively related to trait mindfulness (e.g. [21]).

Regarding psychological flexibility, previous studies have focused on experiential avoidance, which is a process aimed at trying to facilitate psychological flexibility. Experiential avoidance is defined as a “phenomenon that occurs when a person is unwilling to remain in contact with particular private experiences (e.g., bodily sensations, emotions, thoughts, memories, behavioural predispositions) and takes steps to alter the form or frequency of these events and the contexts that occasion them” ([22], p. 1154). For instance, Farr et al. [23] found that experiential avoidance was a significant predictor of psychological distress, this positive association being moderated by self-compassion. Specifically, these authors showed that individuals high in self-compassion reported lower levels of depressive symptoms across low to high experiential avoidance levels than individuals with low self-compassion levels [23]. In addition, Chou et al. [24] found in a sample of 500 university students that experiential avoidance was a significant predictor of Internet addiction, significant depression, and suicidality. Finally, experiential avoidance and self-compassion are strongly and negatively associated [25]. Similarly, mindfulness has been found to negatively correlate with experiential avoidance [26].”

More emphasis needs to be placed on the clinical implications of the findings. The discussion mentions what components should be included when designing an intervention for this group however, the authors do not consider or discuss factors that might not make these interventions feasible. Rather, the authors should consider adjustments that can be made through the university to make uptake of an intervention more likely.

Answer: Following reviewer’s suggestion we have improved the discussion section. Concretely, we have added suggestions for improving intervention in university students taking into account our results:

(Page 17) “Thus, perceived stress in our sample was positively associated with experiential avoidance, which could be regarded as a psychological variable with potential risk. In contrast, perceived stress was negatively correlated with self-compassion and mindfulness, which, in turn, could be seen as protective factors.” 

(Page 19) “Up to now, mindfulness-based interventions for university students have shown promising effectiveness [8,57], though our results suggest that comprehensive programmes more oriented at fostering other crucial process variables for stress reduction such as experiential avoidance and self-compassion are also needed in students showing greater levels of perceived stress. Therefore, these variables should be considered as targets in programmes aimed at fostering wellbeing and mental health in university students. Programmes aimed at increasing certain psychological skills such as Acceptance and Commitment Therapy (ACT; [58,59]) or the Mindful self-compassion programme [60] should be considered when choosing interventions for this population.”

(Pages 21-22): “Thus, our findings underpin the need for formative interventions that would aim firstly to reduce experiential avoidance levels, secondly to promote self-compassion, and thirdly to use mindfulness training as a psychotherapeutic aid. Adequate efforts should be made by higher education institutions to recognise and address university students’ mental health challenges and towards facilitating their mental health recovery, by reducing the long-term consequences of the pandemic on mental health. Thus, these three variables that can be trained and modified could be included as key factors in specific psychological interventions targeted at university students, as previous studies already suggested.”

Please, excuse our modesty when informing about the clinical and practical implications of our study. In fact, it is our view that no single study can be used to inform clinical, policy, prevention, or practice implications. Each study provides suggestive evidence, but whether the findings of the study can be replicated is questionable, especially given the limitations of the study. Thus, the findings of no one study should be used to tell readers what they ought to be doing based on that study. The point is, how can one study with methodological limitations provide directions for clinical work or practice when there is a whole branch of psychology dealing with evidence-based research to inform treatments?

Issues with data collection that need better context:

• Ethnicity of participants not reported – there can be cultural differences influencing certain variables.

Answer: All participants were White; this detail has been included in the participants section (page 6): “The study sample was composed of 589 university teaching students from the San Vicente Mártir Catholic University of Valencia (Spain), of whom 81.2% were female, and the mean age was 22.13 years old (SD = 3.90; range: 18-48). All participants were White.”

• Data was collected on how much family support participants had. How exactly was this measured as it is likely that this was entirely subjective.

Answer: It refers to perceived family support; this detail has been included in the instruments section (page 8): 

“The following sociodemographic characteristics were recorded using an ad hoc questionnaire (response options): age, gender (male/female), having a partner (yes/no), number of children, perceived presence of family support (insufficient/good/very good), employment (yes/no), and emancipation status (yes/no). Likewise, the following academic variables were recorded: having a scholarship (yes/no), which academic year the student was in, total study hours per week, and number of failed subjects over the previous examination period. Along with this survey, participants completed the aspects described below as part of a pencil-and-paper or online set of measurements.”

The following limitation was added too (page 21):

“Fourthly, an ad hoc variable was created to evaluate subjective family support, and further studies should also explore this construct with validated questionnaires.”

• Previous clinical and medical history of participants not reported and nor has this been addressed.

• Authors do not address that data was collected at one time point only, this time point was most likely when students have exams and deadlines which would have influenced findings.

Answer: Following reviewer’s suggestion, those aspects have been included as limitations (page 20-21):

“In this regard, several variables that may have had an influence on the relationship between the outcomes were not assessed. For instance, the role of the previous clinical and medical history of participants or personality traits on perceived stress are not computed. Another example is that data was collected at only one point in time, when students have exams and deadlines which may have influenced findings.”

---

## [Decision Letter · Decision Letter 1]

15 Sep 2022

PONE-D-21-35822R1How mindfulness, self-compassion, and experiential avoidance are related to perceived stress in a sample of university studentsPLOS ONE

Dear Dr. Navarrete,

Thank you for submitting your manuscript to PLOS ONE. After careful consideration, we feel that it has merit but does not fully meet PLOS ONE’s publication criteria as it currently stands. Therefore, we invite you to submit a revised version of the manuscript that addresses the points raised during the review process.

We look forward to receiving your revised manuscript.

Kind regards,

Natalie J. Shook

Academic Editor

PLOS ONE

Journal Requirements:

Reviewers' comments:

Reviewer's Responses to Questions

**Comments to the Author**

1. If the authors have adequately addressed your comments raised in a previous round of review and you feel that this manuscript is now acceptable for publication, you may indicate that here to bypass the “Comments to the Author” section, enter your conflict of interest statement in the “Confidential to Editor” section, and submit your "Accept" recommendation.

Reviewer #1: (No Response)

Reviewer #2: All comments have been addressed

Reviewer #3: All comments have been addressed

2. Is the manuscript technically sound, and do the data support the conclusions?

Reviewer #1: Partly

Reviewer #2: Partly

Reviewer #3: Yes

3. Has the statistical analysis been performed appropriately and rigorously? 

Reviewer #1: Yes

Reviewer #2: Yes

Reviewer #3: Yes

4. Have the authors made all data underlying the findings in their manuscript fully available?

Reviewer #1: Yes

Reviewer #2: No

Reviewer #3: Yes

5. Is the manuscript presented in an intelligible fashion and written in standard English?

Reviewer #1: No

Reviewer #2: Yes

Reviewer #3: Yes

6. Review Comments to the Author

Reviewer #1: The reviewer is able to spot many clarifications in the text. But there are still concerns regarding several areas of the manuscript.

Introduction

I appreciate the authors effort in addressing my prior concerns regarding the introduction. The revised introduction is much more relevant to their study aims. However, I still have significant concerns.

The first paragraph mainly focuses on depression and suicide, which are not the main aims of the present study. I suggest that the authors reframe this paragraph to focus on the role of perceived stress specifically.

The introduction could benefit from a richer discussion of each of the three main constructs: mindfulness, self-compassion, experiential avoidance. Each of these three paragraphs feels underdeveloped. Also, these paragraphs read more as a list of studies that have been conducted, rather than a story about 1) what the construct is, 2) why/how it would be related to stress, and 3) what prior research has shown.

The authors mention a study by Martínez-Rubio that is similar to the present study conducted. To clarify this further, the authors should specify how psychological flexibility was assessed in prior work and which factors were most strongly associated with burnout.

The brief paragraph on sociodemographics and stress does not flow with the prior paragraphs and only mentions a few variables, whereas the present study accounts for several of these variables. In reading it over, it may not be necessary to include a paragraph in the introduction discussing these associations.

Methods

The authors state that they informed participants of the true purpose of the study. Are they concerned that this may have influenced the way that students responded? What are the implications of this for their findings?

Results

I found the data analysis section difficult to follow as written. I suggest that the authors rework this section for clarity.

The authors should mention what variables are significant in step 1 of the regression.

Is it necessary to have 14 demographic/academic covariates entered into the regression model? Did the authors consider running univariable analyses to identify which variables were significantly related to stress, and then entering only those variables into the hierarchical regression analysis?

Discussion

Overall, the discussion section is underdeveloped. The connection between the present study findings and prior work is limited. The importance of the present findings needs to be emphasized further, including the clinical implications paragraph. The degree to which these findings contribute new knowledge to the literature is not clear. The contribution made by this study to the understanding of stress in general is not well articulated. The discussion would be improved by discussing the importance and relationships among these constructs.

Reviewer #2: The revised manuscript is much improved. Many thanks to the authors for the very well done, hard work. A few additional comments. Regarding the question of data availability, the authors have linked to an OSF, but I was not able to access without requesting, which does not meet the public repository standard. If this is incorrect, my apologies and my misunderstanding.

Second, the introduction is much improved and more thorough. I still think a unifying framework to contextualize these three areas (mindfulness, self-compassion, experiential avoidance) as falling under the framework of "how we respond to difficult emotional experiences" or something of the sort would be useful. With college students, we can envision many opportunities for difficult emotions (and of course the literature supports increased risk for distress, depression, etc.), but what you all are focusing on is specifically how people respond to these internal experiences (thoughts, emotions, urges, sensations, memories, etc.). This will help to unite the three processes. You did a good job addressing the comment regarding discussion of the other literature citing how these processes are related.

In your method, your description of the measures is much clearer - thank you.

In the results, the hierarchical regression is well executed and much clearer than the prior analysis. One question that remains is the inclusion of so many independent variables creating a possibility for Type II error. One option is to use a correction of some sort that reduces your p-value threshold for significance. With more than 15 predictor variables, this may be an issue that should be addressed. At minimum, this might be acknowledged as a limitation. One way that I have seen this addressed is to preliminarily examine variables as related to the outcome (PSQ in your case) with t-tests, chi-square, or correlations. If they are significantly related, they are allowed entry into the hierarchical regression. If they are not, they are not included. This is an empirical way to justify inclusion of covariates. Nevertheless, you may wish to keep all variables in the model for theoretical reasons, which is fair. In that case, I would suggest correcting (e.g., Bonferroni) to prevent Type II error.

Finally, the discussion is much more integrated with the literature, and the clinical implications section is improved. One note for improvement -- you discuss experiential avoidance and self-compassion as additional strategies beyond mindfulness that may be useful for college students. You might note the importance of the behavioral components of interventions (like ACT or mindful self-compassion) as an addition to mindfulness. Given that self-compassion and avoidance (or acceptance) are behavioral in nature, this may be a critical addition to mindfulness-focused interventions.

Thanks for the opportunity to review your work, and thanks for your thoughtful and thorough revisions.

Reviewer #3: Thank you to the authors for responding my queries. All my comments and concerns have now been addressed. I don't have any further comments.

7. PLOS authors have the option to publish the peer review history of their article (what does this mean?). If published, this will include your full peer review and any attached files.

Reviewer #1: No

Reviewer #2: No

Reviewer #3: No

---

## [Author Response · Author response to Decision Letter 1]

30 Oct 2022

Journal requirements

Answer: We have reviewed the reference list once again and we have not detected retracted papers. New references have been added as a result of the review process.

Reviewer #1

Authors: We would like to thank you very much for your kind and thoughtful comments. We provide an answer after each comment below.

Comment 1. The first paragraph mainly focuses on depression and suicide, which are not the main aims of the present study. I suggest that the authors reframe this paragraph to focus on the role of perceived stress specifically.

Answer: Following reviewer’s suggestion, we have focused on perceived stress in the first paragraph. We have kept the general comment on the mental health in university students in the two first sentences and removed the part about other mental health problems. Thus, the paragraph continuous with brief and specific information about perceived stress in this population (page 2, paragraphs 1-2):

“The mental health of university students has become a worldwide concern [1]. It is estimated that around 35% of university students meet the diagnostic criteria for a mental disorder [2]. In fact, their risk for developing a mental disorder has significantly increased up to around 50% as a result of the COVID-19 pandemic [3,4]. Overall, university students are particularly vulnerable to stress, which in this population is strongly related to academic underperformance, i.e., failure to fulfil academic obligations, and problematic health behaviors, such as substance use [5,6]. In addition, higher perceived stress levels in university students are associated with poorer quality of life, well-being, and sleep quality [7].

Given the high prevalence of stress among university students, there has been a burgeoning interest in detecting potential protective skills that could be enhanced to improve their mental health, such as mindfulness, self-compassion, and psychological flexibility [8,9]. Many studies have consistently indicated the negative association of mindfulness, compassion, and psychological flexibility with perceived stress in general populations [10,11,12]. Below is a brief conceptualization of them and a summary of the research studying its association with perceived stress among university students.”

Comment 2. The introduction could benefit from a richer discussion of each of the three main constructs: mindfulness, self-compassion, experiential avoidance. Each of these three paragraphs feels underdeveloped. Also, these paragraphs read more as a list of studies that have been conducted, rather than a story about 1) what the construct is, 2) why/how it would be related to stress, and 3) what prior research has shown.

Answer: Thanks for this comment. Following reviewer’s suggestion, we have reframed the introduction section to offer a better description of these constructs. Specifically, we have discussed each one in separate paragraphs and described what they are, how there are linked to perceived stress, and the results of previous studies focused on their relation to perceived stress in university students (see pages 2-4):

“Mindfulness can be defined as a two-component mood that involves the self-regulation of attention, i.e., paying attention to the immediate experience (including internal experiences such as thoughts), and a curious, open, and accepting orientation towards it [13]. The cultivation of mindfulness skills leads to improvements in attentional control, acceptance of one’s experience, and non-reactivity to acute stressors [14,15]. Thus, high-level mindfulness individuals frequently respond to acute stressors with non-reactive acceptance instead of showing a pattern of excessive and blunted responses that could lead to a host of poor health outcomes [16]. This is reflected in previous studies involving university students. For instance, Burger et al. [17] investigated the associations between psychological distress and mindfulness in a sample of 174 university students and found that higher levels of mindfulness (especially the facet of acting with awareness) significantly predicted lower levels of negative affect, fatigue, nervousness, and agitation. Similarly, Muro et al. [18] found that mindfulness was associated with higher levels of life satisfaction in a sample of Spanish university students, a result which is consistent with the multicentre cross-sectional study of Salvarani et al. [19], who found that undergraduate nursing students (n = 622) with higher dispositional mindfulness scores had lower levels of stress symptoms. 

Compassion is a cognitive, affective, and behavioural process that involves recognising suffering, common humanity, empathy, tolerance for uncomfortable feelings, and motivation to act or acting to alleviate the suffering of others and/or oneself [20]. Being self-compassionate during stressful life circumstances enables a pool of emotion-regulation strategies (i.e., healthy reappraisals, emotional acceptance, and self-soothing) that protects against distress [21]. In this line, Rahmandani et al. [22] found a significant negative association between self-compassion and several dimensions of distress (e.g., loss of confidence, anxiety), suggesting that high self-compassion might explain a lower level of stress in university students. In addition, Chan et al. [23] have found in a sample of 536 university students that compassion for others and self-compassion were linked to lower levels of perceived stress. 

Finally, psychological flexibility is defined as the ability to consciously make contact with the present moment and behave in a way that serves valued goals [24]. Previous research about this construct have focused on experiential avoidance instead, which is a process aimed at trying to facilitate psychological flexibility. Experiential avoidance is defined as a “phenomenon that occurs when a person is unwilling to remain in contact with particular private experiences (e.g., bodily sensations, emotions, thoughts, memories, behavioural predispositions) and takes steps to alter the form or frequency of these events and the contexts that occasion them” ([24], p. 1154). Individuals who tend to rigidly avoid uncomfortable or unwanted internal experiences are more vulnerable to stress because this avoidant response pattern inhibits effective behaviors to cope with stress and might even increase the frequency of these unwanted internal experiences [25]. For instance, Chou et al. [26] found in a sample of 500 university students that experiential avoidance was a significant predictor of Internet addiction, significant depression, and suicidality. Moreover, Farr et al. [27] found that experiential avoidance was a significant predictor of depression, anxiety, and stress, this positive association being moderated by self-compassion. Specifically, these authors showed that individuals high in self-compassion reported lower levels of depressive symptoms across low to high experiential avoidance levels than individuals with low self-compassion levels [27].”

Comment 3. The authors mention a study by Martínez-Rubio that is similar to the present study conducted. To clarify this further, the authors should specify how psychological flexibility was assessed in prior work and which factors were most strongly associated with burnout.

Answer: Thanks for this comment. As in the present study, in Martínez-Rubio et al. (2021) experiential avoidance was measured instead of psychological flexibility. This has been indicated in the revised version of the paper, as well as which factors were most strongly associated with burnout (see page 5):

“In this sense, Martínez-Rubio et al. [33] showed that experiential avoidance, self-compassion and the mindfulness facets of observing and acting with awareness were significantly related to lower burnout symptom (‘overload’, ‘lack of development’, and ‘neglect’) levels in a sample of 644 undergraduate nursing and psychology students, but the potential influence of sociodemographic and academic factors in that association was not controlled. Results showed that experiential avoidance was the strongest predictor of the burnout dimensions of ‘overload’ and ‘lack of development’, meanwhile the dimension of ‘neglect’ was better explained by acting with awareness [33].”

Comment 4. The brief paragraph on sociodemographics and stress does not flow with the prior paragraphs and only mentions a few variables, whereas the present study accounts for several of these variables. In reading it over, it may not be necessary to include a paragraph in the introduction discussing these associations.

Answer: Following reviewer’s suggestion, we have now removed the paragraph on sociodemographics.

Comment 5. The authors state that they informed participants of the true purpose of the study. Are they concerned that this may have influenced the way that students responded? What are the implications of this for their findings?

Answer: Our local ethical committee does not allow to mask the aim of the studies to participants. Following the reviewer’s suggestion, we have included this potential bias in the limitations section (page 23) as follows:

“These findings should be interpreted with caution regarding the following limitations. Firstly, […] Finally, participants were aware of the purpose and the general aim of the study, which might have impacted on the results due to social desirability biases.”

Comment 6. I found the data analysis section difficult to follow as written. I suggest that the authors rework this section for clarity.

Answer: Fixed. For clarity, we have tried to improve the readability of this section. Nevertheless, it is important to keep in mind that Plos One emphasizes the need to “report statistical methods in sufficient detail for others to replicate the analysis performed”. Therefore, in line with the editorial policy, we are being forced to maintain all important details, as simplifying the explanation of the hierarchical multiple regression model while keeping all details is not an easy task. 

Thus, we have reordered the information and place quotation marks in the sociodemographic variables. The first paragraph explains descriptive and internal consistency analyses. The second paragraph introduces the hierarchical multiple regression model. Then, we describe some specifications of the regression indices. Finally, we mention the supplementary analyses. Changes can be seen on page XX (Data Analysis section): 

“Data analysis

All data analyses were carried out using the IBM Statistical Package for the Social Sciences (SPSS) v26, Chicago, IL. A descriptive analysis of participant characteristics was carried out using means (M) and standard deviations (SD), for the continuous variables, and frequencies (n) and percentages (%), for the categorical variables. The internal consistency of the instruments was established by calculating Cronbach’s alpha (α). Coefficients above 0.70 were considered adequate [41]. 

A hierarchical multiple regression was carried out to assess the ability of FFMQ-SF (mindfulness), SCS-SF (self-compassion), and AAQ-II (experiential avoidance) measurements to explain PSQ-SF (perceived stress) scores after controlling for the influence of sociodemographic and academic variables. Preliminary analyses were conducted to check the assumptions of multicollinearity, outliers, normality, linearity, and homoscedasticity (S1 and S2 Tables and S3 Fig.), with no major violations noted. First, all categorical sociodemographic and academic variables were recoded as dummy variables as follows: ‘gender’ (0 = male, 1 = female), ‘having a partner’ (0 = no, 1 = yes), ‘children’ (0 = no, 1 = yes), ‘perceived presence of family support’ (0 = no = insufficient, 1 = yes = good/very good), ‘employment’ (0 = no, 1 = yes), ‘left home’ (0 = no, 1 = yes), ‘having a scholarship’ (0 = no, 1 = yes), ‘which academic year the student was in’ (each academic year recoded as dummy variable; 0 = no, 1 = yes). These categorical variables along with ‘age’, ‘total study hours per week’, and ‘number of failed subjects over the previous examination period’ were entered at Step 1. Additionally, FFMQ-SF, SCS-SF, and AAQ-II scores were entered at Step 2. 

Given the large number of predictors included in the hierarchical multiple regression, we adjusted for multiple comparisons using the Benjamini–Hochberg procedure [42] utilizing a false discovery rate of 0.05. Pairwise deletion was the technique used to handle missing data. Semi-partial correlation coefficients (sr) were examined to get an indication of the unique contribution of each variable to the total R square. Specifically, if this coefficient is squared, the percentage of unique variance explained by an independent variable in the dependent variable is obtained [43]. 

Continuing along the path laid down by previous studies [44], the hierarchical multiple regression was re-tested including an ad hoc SCS-SF total score for which the mindfulness and overidentification scores were removed (S4 Table). This analysis was performed due to the partial overlap between trait mindfulness and self-compassion measurements (i.e., FFMQ-SF and SCS-SF, respectively; [32]).”

Comment 7. The authors should mention what variables are significant in step 1 of the regression.

Answer: Thanks for this comment. Following reviewer’s suggestion, we have mentioned which variables were significant in step 1 of the regression (page 16-17):

“Being a woman (sr = 0.16, p < .001), perceived family support (sr = -0.21, p < .001), left home (sr = 0.08, p = .043), and study hours per week (sr = 0.12, p = .004) were significant predictors at Step 1. In the final model, among the psychological measurements, only SCS-SF and AAQ-II significantly explained the PSQ-SF scores, with the AAQ-II recording a higher semi-partial correlation value (sr = 0.31, p < .001) than the SCS (sr = -0.17, p < .001). That is, the AAQ-II and SCS-SF scores uniquely explained around 10% and 3%, respectively, of the variance in PSQ-SF scores. In addition, perceived family support (sr = -0.16, p < .001), study hours per week (sr = 0.12, p < .001), having a partner (sr = 0.09, p = .004), being female (sr = 0.08, p = .010), and age (sr = 0.07, p = .015) made a unique statistically significant contribution to explaining PSQ-SF scores. When the Benjamini–Hochberg correction was applied to correct for multiple comparisons, all significant effects remained significant.”

Regarding the supplementary analysis, this information can be found in Table S4.

Comment 8. Is it necessary to have 14 demographic/academic covariates entered into the regression model? Did the authors consider running univariable analyses to identify which variables were significantly related to stress, and then entering only those variables into the hierarchical regression analysis?

Answer: Our decision was to introduce variables in the model that were theoretically relevant and not rely entirely on a data-driven approach. In fact, our main purpose was to quantify the explanatory power of mindfulness, self-compassion, and experiential avoidance after controlling for the sociodemographic and academic profile of the students. With this purpose in mind, we recruited enough sample to introduce the 17 variables in the model. According to Tabachnick and Fidell (2013), a sample of 186 participants would be the minimum required to include 17 variables in a multiple regression analysis [N > 50 + 8m (where m = number of independent variables)].

Tabachnick, B. G., & Fidell, L. S. (2013). (6th ed.). Boston: Pearson Education.

Even so, we have performed a Benjamini-Hochberg correction for multiple comparisons to decrease the false discovery rate and all significant effects remained significant (see Data analysis, page 12; and Results, page 17):

“Given the large number of predictors included in the hierarchical multiple regression, we adjusted for multiple comparisons using the Benjamini–Hochberg procedure [42] utilizing a false discovery rate of 0.05.”

“Being a woman (sr = 0.16, p < .001), perceived family support (sr = -0.21, p < .001), left home (sr = 0.08, p = .043), and study hours per week (sr = 0.12, p = .004) were significant predictors at Step 1. In the final model, among the psychological measurements, only SCS-SF and AAQ-II significantly explained the PSQ-SF scores, with the AAQ-II recording a higher semi-partial correlation value (sr = 0.31, p < .001) than the SCS (sr = -0.17, p < .001). That is, the AAQ-II and SCS-SF scores uniquely explained around 10% and 3%, respectively, of the variance in PSQ-SF scores. In addition, perceived family support (sr = -0.16, p < .001), study hours per week (sr = 0.12, p < .001), having a partner (sr = 0.09, p = .004), being female (sr = 0.08, p = .010), and age (sr = 0.07, p = .015) made a unique statistically significant contribution to explaining PSQ-SF scores. When the Benjamini–Hochberg correction was applied to correct for multiple comparisons, all significant effects remained significant.

 When the hierarchical multiple regression was re-tested with the SCS-SF total score calculated without adding the mindfulness and overidentification items scores (S4 Table), FFMQ-SF did significantly explain PSQ-SF scores (sr = -0.07; p = .015), along with the aforementioned variables. When the Benjamini–Hochberg correction was applied to correct for multiple comparisons, all significant effects remained significant except for age (adjusted p =.056). In particular, AAQ-II, SCS-SF, and FFMQ-SF scores uniquely explained around 11%, 2%, and 0.55%, respectively, of the variance in PSQ-SF scores.”

Comment 9. Overall, the discussion section is underdeveloped. The connection between the present study findings and prior work is limited. The importance of the present findings needs to be emphasized further, including the clinical implications paragraph. The degree to which these findings contribute new knowledge to the literature is not clear. The contribution made by this study to the understanding of stress in general is not well articulated. The discussion would be improved by discussing the importance and relationships among these constructs.

Answer: Following the reviewer’s suggestion, we have improved the discussion section by highlighting the meaning and implication of the present results. Please, see changes thorough the discussion section (pages 17-24):

“This study examined the explanatory power of mindfulness, self-compassion, and experiential avoidance with regards to perceived stress in a sample of university students while controlling for the influence of relevant sociodemographic and academic variables. When all of them were taken into consideration as potential explanatory variables of perceived stress in the regression model, mindfulness, self-compassion, and experiential avoidance explained a greater variance of perceived stress than all sociodemographic and academic variables together. 

In particular, experiential avoidance was by far the most explanatory variable of levels of perceived stress (direct association). Self-compassion was also a significant explanatory variable that was inversely associated with perceived stress, but to a lower degree. Finally, trait mindfulness showed marginal additive explanatory power in the regression, but when self-compassion scores did not account for mindfulness and overidentification items, mindfulness was shown as a significant explanatory variable inversely associated with perceived stress. Similarly, experiential avoidance significantly explained the levels ‘overload’ and ‘lack of development’ burnout dimensions to a greater degree than self-compassion and mindfulness in our previous study [33]. Thus, both results may be further evidence of a stronger connection between psychopathology and experiential avoidance than between psychopathology and other third-waves constructs (e.g., mindfulness or compassion) in university students.

In our sample, perceived stress was positively associated with experiential avoidance, which could be regarded as a psychological variable with potential risk. In contrast, perceived stress was negatively correlated with self-compassion and mindfulness, which, in turn, could be seen as protective factors. The same scenario was shown regarding the association between experiential avoidance, mindfulness, and self-compassion with burnout dimensions in our previous study [33]. These results can be explained taking the emotion dysregulation model of psychological distress into account [45], which posits that the ways individuals experience and respond to emotions can lead or not to psychological distress. According to this model, motion dysregulation (or misregulation) can occur when the selected emotion regulation strategies do not match the situation or keep the person in an endless struggle to free himself of the unwanted emotions, being experiential avoidance the flagship example among maladaptive strategies. In fact, it is considered a strong transdiagnostic predictor of psychological distress [12]. In contrast, higher mindfulness and self-compassion levels (as a state or trait) are typically related to less psychological distress. For instance, it has been found that low-level mindfulness individuals frequently respond to acute stressors showing a pattern of excessive and blunted responses that could lead to a host of poor health outcomes [16]. In addition, the negative components of self-compassion showed greater associations with psychological distress than the positive counterparts in a recent meta-analytic study [46].

Our findings provide further evidence on the association between experiential avoidance and psychological disturbances [47] and would indicate that stressed students present a tendency to escape or avoid private events (e.g., emotions or memories), regardless of their sociodemographic and academic profile. In this line, previous studies have reported significant associations between experiential avoidance and depression or suicidality in Taiwanese university students [26], as well as between experiential avoidance and burnout syndrome in Spanish undergraduate nursing students [33]. Experiential avoidance would also seem to moderate the positive relationship between anxiety sensitivity and perceived stress in a large community sample of university students from the United States [48]. It is also an important construct in anxiety disorders [49]. In fact, there is a certain amount of anxiety research that demonstrates the mental health costs associated with inflexible development of an avoidance response style, such as the permanence of the experience of bodily arousal in people with panic disorder, being concerned about openly exposing and communicating intense emotional experiences to other people, and the fear of strong emotional impulses in anxiety disorders. The abovementioned mental health costs associated with inflexible psychological features perpetuate anxiety disorders and are related to an experiential avoidance style [49]. 

Self-compassion was the second-best explanatory variable in explaining perceived stress in our sample. Its significant association with perceived stress would indicate that stressed students tend to be self-critical rather than self-kind against failure and adversity, regardless of their sociodemographic and academic profile. These findings are also consistent with previous studies on university students [50,51], in which self-compassion mediated the negative relationship between perceived stress and anxiety and depression symptoms in a large sample of German university students. Furthermore, it was suggested that self-compassion could reduce anxiety and depression levels [50,51], and may increase students’ capacity for managing the emotional demands of their studies [52]. 

Even though the inverse association between dispositional mindfulness and psychopathological symptoms has been consistently reported [53], our results showed a marginally significant relationship between mindfulness as a trait and perceived stress. One possible explanation for that unexpected result was the partial overlap between mindfulness (FFMQ) and self-compassion (SCS) measurements [50]. That is, Neff’s model [31] includes mindfulness (vs. overidentification) as one of the three positive components of self-compassion, which is mirrored in the SCS. In fact, when the overlap between these measurements was solved, mindfulness and perceived stress association were found to be statistically significant. Therefore, the marginal association that was initially found was mostly related to psychometric issues that future research should address using other compassion measurements such as the Sussex-Oxford Compassion Scales [28]. Thus, our results would indicate that stressed students tend to be unaware of the present moment experience and to respond reactively to it. In this line, previous studies have shown the inverse association between mindfulness and perceived stress levels [17, 19] as well as the positive association between mindfulness and satisfaction with life [18].

Regarding sociodemographic and academic variables, high total study hours per week, having a partner (vs. being single), being female (vs. being male), and being older were significantly associated with high levels of perceived stress. In addition, perceived family support was found to be a protective variable, a good/very good perception of family support being associated with lower levels of perceived stress. Finally, having children, being in employment, having left the family home, having a scholarship, the academic year, and the number of failed subjects were not significant contributors to higher levels of perceived stress.

As mentioned, higher perceived family support significantly explained lower perceived stress levels in our sample of university students. These findings may provide a basis for explaining how family support may be a relevant buffering variable in the perception of the stress that occurs among undergraduate students [54,55]. For example, in a sample of Chinese university students, family cohesion explained social adjustment by increasing a sense of security [55], altogether psychosocial factors that are known to buffer the harmful effects of stressors on well-being. Conversely, having a partner was positively associated with perceived stress. It is possible that the family is more supportive of difficulties during the university experience than a partner, who may be more of an obstacle or a source of tension during this period.

Regarding the significant positive association found between perceived stress and study hours per week, the perception of overload due to the amount of study hours has previously been associated with stress and also burnout [56,57]. Furthermore, the load of study hours per week constitutes not only an associated perceived stress variable, but also an explanatory factor that could impair academic performance among university students [58]. Finally, as in other studies, being female and of an older age also explained increased perceived stress levels [35,59,60].

Up to now, mindfulness-based interventions for university students have shown promising effectiveness [8,61], though our results suggest that comprehensive programmes more oriented at fostering other crucial process variables for stress reduction such as experiential avoidance and self-compassion are also needed in students showing greater levels of perceived stress. Therefore, these variables should be considered as targets in programs aimed at fostering well-being and mental health in university students. Programs aimed at increasing certain psychological skills such as Acceptance and Commitment Therapy [62,63] or the Mindful self-compassion program [64] should be considered when choosing interventions for this population. Indeed, it should be noted the importance of the behavioral components of these interventions as an addition to mindfulness. Given that self-compassion and avoidance (or acceptance) are behavioral in nature, this may be a critical addition to mindfulness-based interventions.

Finally, taking into account that our sample was composed of university teaching students it should be noted that psychological flexibility and self-compassion are important aspects of social and emotional competences in teaching professionals’ performance. In turn, the social and emotional competence of teachers is a key point to promote healthy student-teacher relationships, effective classroom management, and effective social and emotional learning implementation that contributes to a healthy classroom atmosphere. This healthy classroom atmosphere benefits students’ social, emotional and academic learning process which, in turn, has an indirect effect on teachers’ social/emotional well-being [65].”

Reviewer #2

Comment 1. The revised manuscript is much improved. Many thanks to the authors for the very well done, hard work. A few additional comments. Regarding the question of data availability, the authors have linked to an OSF, but I was not able to access without requesting, which does not meet the public repository standard. If this is incorrect, my apologies and my misunderstanding.

Answer: Sorry, the person from our team who uploaded the database has had technical problems with his OSF account. We think that this can be related to the impossibility to download the database. We have uploaded the database again, here is the new link: 

https://osf.io/qjp5h/

Comment 2. Second, the introduction is much improved and more thorough. I still think a unifying framework to contextualize these three areas (mindfulness, self-compassion, experiential avoidance) as falling under the framework of "how we respond to difficult emotional experiences" or something of the sort would be useful. With college students, we can envision many opportunities for difficult emotions (and of course the literature supports increased risk for distress, depression, etc.), but what you all are focusing on is specifically how people respond to these internal experiences (thoughts, emotions, urges, sensations, memories, etc.). This will help to unite the three processes. You did a good job addressing the comment regarding discussion of the other literature citing how these processes are related. In your method, your description of the measures is much clearer - thank you.

Answer: Thanks for this comment. Following reviewer’s suggestion, we have emphasized in the introduction section the unifying framework of the three constructs we study here in relation to perceived stress in university students (see page 5):

“Overall, these are three psychological constructs of “third wave” cognitive behavioral therapies, which focus on the relationship between the individual and his/her thoughts and emotions regardless of the content [12]. As mentioned, university students show increased risk for distress and focusing on specifically how they respond to the internal experiences (thoughts, emotions, urges, sensations, memories, etc.) associated with their stressful circumstances might provide further insights about the relevance of applying third-wave approaches (e.g., mindfulness-, compassion-based interventions or acceptance and commitment therapy). As a matter of fact, it seems that increasing mindfulness facets, compassion, or psychological flexibility (by reducing experiential avoidance) may help to improve psychological distress in university students (e.g. [17,19,27]). However, very few studies have simultaneously studied the ability of trait mindfulness, compassion, and experiential avoidance, along with sociodemographic and academic variables, to explain levels of psychological distress in university students. Including all those variables in the same model would add specific information about which of these psychological constructs should be more emphasized in mental health interventions for university students.”

Comment 3. In the results, the hierarchical regression is well executed and much clearer than the prior analysis. One question that remains is the inclusion of so many independent variables creating a possibility for Type II error. One option is to use a correction of some sort that reduces your p-value threshold for significance. With more than 15 predictor variables, this may be an issue that should be addressed. At minimum, this might be acknowledged as a limitation. One way that I have seen this addressed is to preliminarily examine variables as related to the outcome (PSQ in your case) with t-tests, chi-square, or correlations. If they are significantly related, they are allowed entry into the hierarchical regression. If they are not, they are not included. This is an empirical way to justify inclusion of covariates. Nevertheless, you may wish to keep all variables in the model for theoretical reasons, which is fair. In that case, I would suggest correcting (e.g., Bonferroni) to prevent Type II error.

Answer: Thanks for this important comment. As the reviewer says, our decision was to introduce variables in the model that were relevant to control for theoretical reasons. In fact, our main purpose was to quantify the explanatory power of mindfulness, self-compassion, and experiential avoidance after controlling for the sociodemographic and academic profile of the students. Following reviewer’s suggestion, we have performed a Benjamini-Hochberg correction for multiple comparisons to decrease the false discovery rate (Data analysis, page 12; Results, page 16-17):

“Given the large number of predictors included in the hierarchical multiple regression, we adjusted for multiple comparisons using the Benjamini–Hochberg procedure [42] utilizing a false discovery rate of 0.05. “

“Being a woman (sr = 0.16, p < .001), perceived family support (sr = -0.21, p < .001), left home (sr = 0.08, p = .043), and study hours per week (sr = 0.12, p = .004) were significant predictors at Step 1. In the final model, among the psychological measurements, only SCS-SF and AAQ-II significantly explained the PSQ-SF scores, with the AAQ-II recording a higher semi-partial correlation value (sr = 0.31, p < .001) than the SCS (sr = -0.17, p < .001). That is, the AAQ-II and SCS-SF scores uniquely explained around 10% and 3%, respectively, of the variance in PSQ-SF scores. In addition, perceived family support (sr = -0.16, p < .001), study hours per week (sr = 0.12, p < .001), having a partner (sr = 0.09, p = .004), being female (sr = 0.08, p = .010), and age (sr = 0.07, p = .015) made a unique statistically significant contribution to explaining PSQ-SF scores. When the Benjamini–Hochberg correction was applied to correct for multiple comparisons, all significant effects remained significant.

 When the hierarchical multiple regression was re-tested with the SCS-SF total score calculated without adding the mindfulness and overidentification items scores (S4 Table), FFMQ-SF did significantly explain PSQ-SF scores (sr = -0.07; p = .015), along with the aforementioned variables. When the Benjamini–Hochberg correction was applied to correct for multiple comparisons, all significant effects remained significant except for age (adjusted p =.056). In particular, AAQ-II, SCS-SF, and FFMQ-SF scores uniquely explained around 11%, 2%, and 0.55%, respectively, of the variance in PSQ-SF scores.”

Comment 4. Finally, the discussion is much more integrated with the literature, and the clinical implications section is improved. One note for improvement -- you discuss experiential avoidance and self-compassion as additional strategies beyond mindfulness that may be useful for college students. You might note the importance of the behavioral components of interventions (like ACT or mindful self-compassion) as an addition to mindfulness. Given that self-compassion and avoidance (or acceptance) are behavioral in nature, this may be a critical addition to mindfulness-focused interventions.

Answer: Thank you for this important suggestion. We agree with the reviewer, we have included this reflection in the paper as follows (page 21-22): 

“Up to now, mindfulness-based interventions for university students have shown promising effectiveness [8,61], though our results suggest that comprehensive programmes more oriented at fostering other crucial process variables for stress reduction such as experiential avoidance and self-compassion are also needed in students showing greater levels of perceived stress. Therefore, these variables should be considered as targets in programs aimed at fostering well-being and mental health in university students. Programs aimed at increasing certain psychological skills such as Acceptance and Commitment Therapy [62,63] or the Mindful self-compassion program [64] should be considered when choosing interventions for this population. Indeed, it should be noted the importance of the behavioral components of these interventions as an addition to mindfulness. Given that self-compassion and avoidance (or acceptance) are behavioral in nature, this may be a critical addition to mindfulness-based interventions.”

---

## [Editor Report · Decision Letter 2]

23 Nov 2022

PONE-D-21-35822R2How mindfulness, self-compassion, and experiential avoidance are related to perceived stress in a sample of university studentsPLOS ONE

Dear Dr. Navarrete,

Thank you for submitting your manuscript to PLOS ONE. After careful consideration, we feel that it has merit but does not fully meet PLOS ONE’s publication criteria as it currently stands. Therefore, we invite you to submit a revised version of the manuscript that addresses the points raised during the review process.

Thank you for your responsiveness to the reviewers' comments. The manuscript is much improved. There are a few minor edits that I would like you to make before I accept the manuscript. First, please note how sample size or termination of data collection was determined (e.g., a priori power analysis). If an a priori power analysis was not conducted, please report a sensitivity analysis to indicate the minimum effect size that the study is powered to detect. Finally, please be true to null hypothesis testing. An effect is either significant or not significant. Please remove discussion of "marginally significant" effects.  

We look forward to receiving your revised manuscript.

Kind regards,

Natalie J. Shook

Academic Editor

PLOS ONE
---

## [Author Response · Author response to Decision Letter 2]

7 Dec 2022

COMMENT 1. Thank you for your responsiveness to the reviewers' comments. The manuscript is much improved. There are a few minor edits that I would like you to make before I accept the manuscript. First, please note how sample size or termination of data collection was determined (e.g., a priori power analysis). If an a priori power analysis was not conducted, please report a sensitivity analysis to indicate the minimum effect size that the study is powered to detect. 

Authors: A priori power analysis was not conducted. Thus, following the editor’s suggestion, we have reported a sensitivity analysis to indicate the minimum effect size that the study is powered to detect:

“Data analysis

Given the large number of predictors included in the hierarchical multiple regression, we adjusted for multiple comparisons using the Benjamini–Hochberg procedure [42] utilizing a false discovery rate of 0.05. In addition, a sensitivity analysis using GPower v3.1 was conducted to indicate the minimum effect size that the study was powered to detect. Pairwise deletion was the technique used to handle missing data. Semi-partial correlation coefficients (sr) were examined to get an indication of the unique contribution of each variable to the total R square. Specifically, if this coefficient is squared, the percentage of unique variance explained by an independent variable in the dependent variable is obtained [43].”

Results

“Table 2 shows the summary of the results from the hierarchical multiple regression analysis. The complete output is shown in S2 Table. According to the sample size (n = 589) and number of predictors (17 variables), the regression would be sensitive to effects of f2 = 0.03 (small) with 80% power (alpha = .05, two-tailed). Sociodemographic and academic variables were entered at Step 1, explaining 12% of the variance in PSQ-SF scores. Sociodemographic and academic variables were entered at Step 1, explaining 12% of the variance in PSQ-SF scores (f2 = 0.14). After entering the FFMQ-SF, SCS-SF, and AAQ-II scores at Step 2 the total variance explained by the model was 50%, F(17, 542) = 32.29, p < .001, f2 = 1. In other words, the psychological measures (FFMQ-SF, SCS-SF, and AAQ-II) explained an additional 38% in perceived stress levels (PSQ) after controlling for sociodemographic and academic variables, R squared change = .381, F change (3, 542) = 139.41, p < .001.”

COMMENT 2. Finally, please be true to null hypothesis testing. An effect is either significant or not significant. Please remove discussion of "marginally significant" effects.

Authors: Following the editor’s suggestion, we have removed discussion of "marginally significant" effects. Concretely, we have changed "marginally significant effects” for “non-significant effects”. Please, see changes in the Discussion section:

“In particular, experiential avoidance was by far the most explanatory variable of levels of perceived stress (direct association). Self-compassion was also a significant explanatory variable that was inversely associated with perceived stress, but to a lower degree. Finally, trait mindfulness showed a non-significant additive explanatory power in the regression, but when self-compassion scores did not account for mindfulness and overidentification items, mindfulness was shown as a significant explanatory variable inversely associated with perceived stress.”

“Even though the inverse association between dispositional mindfulness and psychopathological symptoms has been consistently reported [53], our results showed a non-significant relationship between mindfulness as a trait and perceived stress. One possible explanation for that unexpected result was the partial overlap between mindfulness (FFMQ) and self-compassion (SCS) measurements [50]. That is, Neff’s model [31] includes mindfulness (vs. overidentification) as one of the three positive components of self-compassion, which is mirrored in the SCS. In fact, when the overlap between these measurements was solved, mindfulness and perceived stress association were found to be statistically significant. Therefore, the non-significant association that was initially found was mostly related to psychometric issues that future research should address using other compassion measurements such as the Sussex-Oxford Compassion Scales [28].”

---

## [Editor Report · Decision Letter 3]

10 Jan 2023

How mindfulness, self-compassion, and experiential avoidance are related to perceived stress in a sample of university students

PONE-D-21-35822R3

Dear Dr. Navarrete,

We’re pleased to inform you that your manuscript has been judged scientifically suitable for publication and will be formally accepted for publication once it meets all outstanding technical requirements.

Kind regards,

Natalie J. Shook

Academic Editor

PLOS ONE
---

## [Editor Report · Acceptance letter]

25 Jan 2023

PONE-D-21-35822R3 

How mindfulness, self-compassion, and experiential avoidance are related to perceived stress in a sample of university students 

Dear Dr. Navarrete:

I'm pleased to inform you that your manuscript has been deemed suitable for publication in PLOS ONE. Congratulations! Your manuscript is now with our production department. 

Kind regards, 

on behalf of

Dr. Natalie J. Shook 

Academic Editor

PLOS ONE